# On the Power of Differentiable Learning versus PAC and SQ Learning

**Emmanuel Abbe**[*]
EPFL

**Pritish Kamath**[*]
Google Research
pritish@ttic.edu

**Eran Malach**[*]
HUJI

**Colin Sandon**[*]
MIT
csandon@mit.edu

**Nathan Srebro**[*]
TTIC
nati@ttic.edu

## Abstract

We study the power of learning via mini-batch stochastic gradient descent (SGD) on the population loss, and batch Gradient Descent (GD) on the empirical loss, of a differentiable model or neural network, and ask what learning problems can be learnt using these paradigms. We show that SGD and GD can always simulate learning with statistical queries (SQ), but their ability to go beyond that depends on the precision $\rho$ of the gradient calculations relative to the minibatch size $b$ (for SGD) and sample size $m$ (for GD). With fine enough precision relative to minibatch size, namely when $b\rho$ is small enough, SGD can go beyond SQ learning and simulate any sample-based learning algorithm and thus its learning power is equivalent to that of PAC learning; this extends prior work that achieved this result for $b = 1$. Similarly, with fine enough precision relative to the sample size $m$, GD can also simulate any sample-based learning algorithm based on $m$ samples. In particular, with polynomially many bits of precision (i.e. when $\rho$ is exponentially small), SGD and GD can both simulate PAC learning regardless of the mini-batch size. On the other hand, when $b\rho^2$ is large enough, the power of SGD is equivalent to that of SQ learning.

## 1 Introduction

A leading paradigm that has become the predominant approach to learning is that of *differentiable learning*, namely using a parametric function class $f_{\boldsymbol{w}}(x)$, and learning by performing mini-**b**atch **S**tochastic **G**radient **D**escent (bSGD) updates (using gradients of the loss on a mini-batch of $b$ independent samples per iteration) or **f**ull-**b**atch **G**radient **D**escent (fbGD) updates (full gradient descent on the empirical loss, using the same $m$ samples in all iterations). Feed-forward neural networks are a particularly popular choice for the parametric model $f_{\boldsymbol{w}}$. One approach to understanding differentiable learning is to think of it as a method for minimizing the empirical error of $f_{\boldsymbol{w}}$ with respect to $\boldsymbol{w}$, i.e. as an empirical risk minimization (ERM). ERM is indeed well understood, and is in a sense a universal learning rule, in that any hypothesis class that is PAC learnable is also learnable using ERM. Furthermore, since poly-sized feed-forward neural networks can represent any poly-time computable function, we can conclude that ERM on neural networks can efficiently learn any tractable problem. But this view of differentiable learning ignores two things.

Firstly, many ERM problems, including ERM on any non-trivial neural network, are highly non-convex and (stochastic) gradient descent might not find the global minimizer. As we just discussed,

---

[*]Authors in alphabetical order.

35th Conference on Neural Information Processing Systems (NeurIPS 2021).

pretending that bSGD or fbGD do find the global minimizer would mean we can learn all poly-time computable functions, which is known to be impossible[2] [e.g. 12, 13]. In fact, even minimizing the empirical risk on a neural net with two hidden units, and even if we assume data is exactly labeled by such a network, is already NP-hard, and cannot be done efficiently with bSGD/fbGD [6]. We already see that bSGD and fbGD are not the same as ERM, and asking "what can be learned by bSGD/fbGD" is quite different from asking "what can be learned by ERM".

Furthermore, bSGD and fbGD might also be more powerful than ERM. Consider using a highly overparametrized function class $f_{\boldsymbol{w}}$, with many more parameters than the number of training examples, as is often the case in modern deep learning. In such a situation there would typically be many empirical risk minimizers (zero training error solutions), and most of them would generalize horribly, and so the ERM principal on its own is not sufficient for learning [18]. Yet we now understand how fbGD and bSGD incorporate intricate implicit bias that leads to particular empirical risk minimizers which might well ensure learning [19, 17].

All this justifies studying differentiable learning as a different and distinct paradigm from empirical risk minimization. The question we ask is therefore: **What can be learned by performing mini-batch stochastic gradient descent, or full-batch gradient descent, on some parametric function class $f_{\boldsymbol{w}}$, and in particular on a feed-forward neural network?** Answering this question is important not only for understanding the limits of what we could possibly expect from differentiable learning, but even more so in guiding us as to how we should study differentiable learning, and ask questions, e.g., about its power relative to kernel methods [e.g. 22, 3, 4, 14, 8–10, 16], or the theoretical benefits of different architectural innovations [e.g. 15].

A significant, and perhaps surprising, advance toward answering this question was recently presented by Abbe and Sandon [1], who showed that bSGD with a single example per iteration (i.e. a minibatch of size 1) can simulate any poly-time learning algorithm, and hence is as powerful as PAC learning. On the other hand, they showed training using population gradients (infinite batch size, or even very large batch sizes) and low-precision (polynomial accuracy, i.e. a logarithmic number of bits of precision), is no more powerful than learning with Statistical Queries (SQ), which is known to be strictly less powerful than PAC learning [11, 7]. This seems to suggest non-stochastic fbGD, or even bSGD with large batch sizes, is not universal in the same way as single-example bSGD. As we will see below, it turns out this negative result depends crucially on the allowed precision, and its relationship to the batch, or sample, size.

**Our Contributions.** In this paper, we take a more refined view of this dichotomy, and consider learning with bSGD with larger mini-batch sizes $b > 1$, as is more typically done in practice, as well as with fbGD. We ask whether the ability to simulate PAC learning is indeed preserved also with larger mini-batch sizes, or even full batch GD? Does this universality rest on using single examples, or perhaps very small mini-batches, or can we simulate any learning algorithm also without such extreme stochasticity? We discover that this depends on the relationship of the batch size $b$ and the precision $\rho$ of the gradient calculations. That is, to understand the power of differentiable learning, we need to also explicitly consider the numeric precision used in the gradient calculations, where $\rho$ is an arbitrary additive error we allow.

We first show that regardless of the mini-batch size, bSGD is always able to simulate any SQ method, and so bSGD is at least as powerful as SQ learning. When the mini-batch size $b$ is large relative to the precision, namely $b = \omega(\log(n)/\rho^2)$, where $n$ is the input dimension and we assume the model size and number of iterations are polynomial in $n$, bSGD is not any more powerful than SQ. But when $b < 1/(8\rho)$, or in other words with fine enough precision $\rho < 1/(8b)$, bSGD can again simulate any sample-based learning method, and is as powerful as PAC learning (the number of SGD iterations and size of the model used depend polynomially on the sample complexity of the method being simulated, but the mini-batch size $b$ and precision $\rho$ do not, and only need to satisfy $b\rho < 1/8$—see formal results in Section 3). We show a similar result for fbGD, with a dependence on the sample size $m$: with low precision (large $\rho$) relative to the sample size $m$, fbGD is no more powerful than SQ. But with fine enough precision relative to the sample size, namely when $\rho < 1/(8m)$, fbGD can again simulate any sample-based learning method based on $m$ samples (formal results are deferred to supplementary material).

---

[2]Subject to mild cryptographic assumptions, such as the existence of one-way functions, i.e. the existence of cryptography itself, and where we are referring to bSGD/fbGD running for polynomially many steps.

We see then, that with fine enough precision, differentiable learning with any mini-batch size, or even using full-batch gradients (no stochasticity in the updates), is as powerful as any sample-based learning method. The required precision does depend on the mini-batch or sample size, but only linearly. That is, the number of *bits of precision* required is only logarithmic in the mini-batch or sample sizes. And with a linear (or even super-logarithmic) number of bits of precision (i.e. $\rho = 2^{-n}$), bSGD and fbGD can simulate arbitrary sample-based methods with any polynomial mini-batch size; this is also the case if we ignore issue of precision and assume exact computation (corresponding to $\rho = 0$).

On the other hand, with low precision (high $\rho$, i.e. only a few bits of precision, which is frequently the case when training deep networks), the mini-batch size $b$ plays an important role, and simulating arbitrary sample based methods is provably *not* possible using fbGD, or with bSGD with a mini-batch size that is too large, namely $b = \omega(\log(n)/\rho^2)$. Overall, except for an intermediate regime between $1/\rho$ and $\log(n)/\rho^2$, we can precisely capture the power of bSGD.

**Computationally Bounded and Unbounded Learning.** Another difference versus the work of Abbe and Sandon is that we discuss both computationally tractable and intractable learning. We show that poly-time PAC and SQ are related, in the sense described above, to bSGD and fbGD on a poly-sized neural network, whereas computationally unbounded PAC and SQ (i.e. limited only by the number of samples or number of statistical queries, but not runtime) are similarly related to bSGD and fbGD on an arbitrary differentiable model $f_w$. In fact, to simulate PAC and SQ, we first construct an arbitrary $f_w$, and then observe that if the PAC or SQ method is poly-time computable, then the computations within it can be expressed as poly-size circuits, which can in turn be simulated as poly-size sub-networks, allowing us to implement $f_w$ as a neural net.

**Answering SQs using Samples.** Our analysis relies on introduction of a variant of SQ learning which we refer to as mini-**b**atch **S**tatistical **Q**ueries (bSQ, and we similarly introduce a full-batch variant, fbSQ). In this variant, which is related to the Honest-SQ model [20, 21], statistical queries are answered using a mini-batch of samples drawn from the source distribution, up to some precision. We first show that bSQ methods can always be simulated by bSGD, by constructing a differentiable model where at each step the derivatives with respect to some of the parameters contain the answers to the statistical queries. We then relate bSQ to SQ and PAC, based on the relationship between the mini-batch size and precision. In order to simulate PAC using bSQ, we develop a novel "sample extraction" that uses mini-batch statistical queries (on independently sampled mini-batches) to extract a single sample drawn from the source distribution. This procedure might be of independent interest, perhaps also in studying privacy, where such an extraction is not desirable. Our study of the relationship of bSQ to SQ and PAC, summarized in Section 4, also sheds light on how well the SQ framework captures learning by answering queries using empirical averages on a sample, which is arguably one of the main motivations for the inverse-polynomial tolerance parameter in the SQ framework.

## 2   Learning Paradigms

We consider learning a predictor $f : \mathcal{X} \to \mathbb{R}$ over an *input space* $\mathcal{X}$, so as to minimize its *population loss* $\mathcal{L}_{\mathcal{D}}(f) := \mathbb{E}_{(x,y)\sim\mathcal{D}} \, \ell(f(x), y)$ with respect to a *source distribution* $\mathcal{D}$ over $\mathcal{X} \times \mathcal{Y}$, where $\ell : \mathbb{R} \times \mathcal{Y} \to \mathbb{R}_{\geq 0}$ is a loss function. Unless noted otherwise, we take the loss function to be the square-loss $\ell_{\mathrm{sq}}(\hat{y}, y) = \frac{1}{2}(\hat{y} - y)^2$. For concreteness, we always take $\mathcal{Y} = \{0, 1\}$ and $\mathcal{X} = \{0, 1\}^n$.

**Differentiable Learning.** We study learning by (approximate) gradient descent on a differentiable model. Formally, a *differentiable model* of size $p$ is a mapping $f : \mathbb{R}^p \times \mathcal{X} \to \mathbb{R}$, denoted $f_w(x)$, where $w \in \mathbb{R}^p$ are the parameters, or "weights", of the model, and for every $x \in \mathcal{X}$ there exists a gradient $\nabla_w f_w(x)$ for almost every $w \in \mathbb{R}^p$ (i.e, outside of a measure-zero set of weights).[3] A notable special case of differentiable models we study is that of feed-forward neural networks.

For a differentiable model $f_w(x)$, an initialization distribution $\mathcal{W}$ over $\mathbb{R}^p$, mini-batch size $b$, gradient precision $\rho$, and stepsize[4] $\gamma$, the mini-**b**atch **S**tochastic **G**radient **D**escent (bSGD) method

---

[3]Allowing $f_w(x)$ to be non-differentiable on a measure zero set might seem overly generous, but it's simpler to state, and a more conservative restriction wouldn't affect our results. Our simulations use either everywhere differentiable models, or neural networks with piece-wise linear activation, resulting in a piece-wise linear models with a finite number of pieces.

[4]The stepsize doesn't play an important role in our analysis. We can also allow variable or adaptive stepsize sequences without changing the results—for simplicity of presentation we stick with a fixed stepsize, and in fact in our constructions just use a fixed constant (not dependent on other parameters) stepsize of $\gamma = 1$ or $\gamma = 2$.

operates by computing iterates of the form: $\boldsymbol{w}^{(0)} \sim \mathcal{W}$ and $\boldsymbol{w}^{(t+1)} \leftarrow \boldsymbol{w}^{(t)} - \gamma g_t$, where $g_t$ is a $\rho$-**approximate rounding** of the mini-batch (empirical) *clipped* gradient $\overline{\nabla}\mathcal{L}_{S_t}(f_{\boldsymbol{w}^{(t)}}) := \frac{1}{b}\sum_{i=1}^{b}[\nabla\ell(f_{\boldsymbol{w}^{(t)}}(x_{t,i}), y_{t,i})]_1$, where the mini-batches $S_t = ((x_{t,1}, y_{t,1}), \ldots, (x_{t,b}, y_{t,b})) \sim \mathcal{D}^b$ containing $b$ samples each are sampled independently at each iteration, and $[\alpha]_1$ denotes entry-wise clipping of $\alpha$ to $[-1, 1]$. We say that $u \in [-1, 1]^p$ is a $\rho$-approximate rounding of $v \in [-1, 1]^p$ if $u \in \rho \cdot \mathbb{Z}^p$ (namely, each entry of $u$ is an integral multiple of $\rho$), and $\|u - v\|_\infty \leq 3\rho/4$. Thus, overall $g_t$ is a valid gradient estimate if it satisfies

$$g_t \in \rho \cdot \mathbb{Z}^p \quad \text{and} \quad \|g_t - \overline{\nabla}\mathcal{L}_{S_t}(f_{\boldsymbol{w}^{(t)}})\|_\infty \leq 3\rho/4 \tag{1}$$

**Precision, Rounding and Clipping.** The clipping and rounding we use captures using $d = -\log\rho$ bits of precision, and indeed we generally consider $\rho = 2^{-d}$ where $d \in \mathbb{N}$.

We consider *clipped* gradients because the precision $\rho$ makes sense only when considered relative to the scale of the gradients. Clipping is an easy way to ensure we do not "cheat" by using very large magnitude gradients to circumvent the precision limit, and can be thought of as a way of handling overflow. We note however that in all our simulation constructions, we always have $\|\nabla\ell(f_{\boldsymbol{w}^{(t)}}(x), y)\|_\infty \leq 1$ for all $\boldsymbol{w}^{(t)}$ and all $(x, y)$, and hence clipping plays no role. We could have alternatively said the method "fails" or is "invalid" if a larger magnitude gradient is encountered.

In our *rounding error* model, for any integer $q$, values in $(q\rho - \frac{\rho}{4}, q\rho + \frac{\rho}{4})$ get rounded to $q\rho$, whereas, values in $[q\rho + \rho/4, q\rho + 3\rho/4]$ can get rounded either to $q\rho$ or $(q + 1)\rho$. This error model is a proxy for rounding errors in real finite precision arithmetic, where we have uncertainty about the least significant bit when representing real numbers in $[-1, 1]$ with $d$ bits of precision, as this final bit represents a rounding of the ignored lower order bits. Viewed another way, a $\rho$-approximate rounding of $v \in [-1, 1]$ can be obtained by considering a $\rho/4$-approximation of $v$, i.e. $\tilde{v} \in [-1, 1]$ s.t. $|v - \tilde{v}| \leq \rho/4$, and then (deterministically) rounding $\tilde{v}$ to the nearest integer multiple of $\rho$.

**Learning with bSGD.** A learning method $\mathcal{A}$ is said to be a bSGD$(T, \rho, b, p, r)$ method if it operates by computing bSGD iterates for some differentiable model $f : \mathbb{R}^p \times \mathcal{X} \to \mathbb{R}$ (i.e., with $p$ parameters), starting from some (randomized) initialization $\mathcal{W}$ which uses $r$ random bits,[5] with some stepsize $\gamma$, given $\rho$-approximate gradients over batches of size $b$; the output of $\mathcal{A}$ is the final iterate $f_{\boldsymbol{w}^{(T)}} : \mathcal{X} \to \mathbb{R}$. We say that $\mathcal{A}$ ensures error $\varepsilon$ on a source distribution $\mathcal{D}$ if $\mathbb{E}[\sup \mathcal{L}_D(f_{\boldsymbol{w}^{(T)}})] \leq \varepsilon$, where the expectation is over both the initialization $\boldsymbol{w}^{(0)}$ and the mini-batches $S_t \sim \mathcal{D}^b$, and the sup is over all gradient estimates $g_t$ satisfying (1) at iterates where $f_{\boldsymbol{w}}$ is differentiable at $\boldsymbol{w}^{(t)}$ for all $x_{t,i} \in S_t$.[6,7] We denote by bSGD$^\sigma_{\mathsf{NN}}(T, \rho, b, p, r)$ the family of bSGD methods where the differentiable model is implemented by a neural network with $p$ parameters, using the poly-time computable[8] activation function $\sigma$ and where the initialization distribution $\mathcal{W}$ can be sampled from in $\mathrm{poly}(n)$ time.

We also consider learning with **f**ull-**b**atch **G**radient **D**escent, i.e. gradient descent on an empirical loss, where the entire training set of size $m$ is used in each iteration. We defer details of this definition and corresponding results to supplementary material.

**PAC and SQ Learning.** A learning method $\mathcal{A}$ is said to be a PAC$(m, r)$ method[9] if it takes in a set of samples $S \subseteq \mathcal{X} \times \mathcal{Y}$ of size $m$, uses $r$ bits of randomness and returns a predictor $f : \mathcal{X} \to \mathbb{R}$. We say that $\mathcal{A}$ ensures error $\varepsilon$ on a source distribution $\mathcal{D}$ if $\mathbb{E}[\mathcal{L}_{\mathcal{D}}(f)] \leq \varepsilon$, where the expectation is over

---

[5]Namely, $\mathcal{W} : \{0, 1\}^r \to \mathbb{R}^p$ such that $\boldsymbol{w}^{(0)} \sim \mathcal{W}(s)$ for $s \sim$ uniform over $\{0, 1\}^r$.

[6]Formally, and for simplicity, if $f_{\boldsymbol{w}}$ is not differentiable at $\boldsymbol{w}^{(t)}$ for some $x_{t,i}$, we consider any $g_t \in [-1, 1]^p$ as valid. But recall that $f_{\boldsymbol{w}}(x)$ is differentiable almost everywhere. We could have required a more conservative behaviour without affecting any of our results. In particular, in all our simulation constructions, $f_{\boldsymbol{w}}(x)$ is *always* differentiable at $\boldsymbol{w}^{(t)}$ encountered.

[7]Defining the error as $\mathbb{E}[\sup \mathcal{L}_D(f_{\boldsymbol{w}^{(T)}})]$ can be interpreted as allowing the rounding errors on $g_t$ to also depend on random samples in future steps. A more conservative definition would involve $\mathbb{E}_{\boldsymbol{w}^{(0)}} \mathbb{E}_{S_1} \sup_{g_1} \ldots \mathbb{E}_{S_T} \sup_{g_T} \mathcal{L}_D(f_{\boldsymbol{w}^{(T)}})$. However, this distinction does not change any of our results and we stick to the simpler $\mathbb{E} \sup$ definition for convenience.

[8]We say that $\sigma : \mathbb{R} \to \mathbb{R}$ is "polynomial-time computable" if there exists a Turing machine that, for any given $z \in \mathbb{R}$ and desired precision, computes $\sigma(z)$ and $\sigma'(z)$ to within the desired precision in time that is polynomial in the bit length of the representation of $z$ and the number of desired bits of precision.

[9]The usage of the acronym PAC here is technically improper as we use it to refer to a *method* and not a class of learning problems. We use it for notational convenience and historical reasons.

$S \sim \mathcal{D}^m$ and the randomness in $\mathcal{A}$. Similarly, we say that $\mathcal{A}$ is a $\mathsf{PAC}_{\mathsf{TM}}(m, r, \mathrm{TIME})$ algorithm if it is a $\mathsf{PAC}(m, r)$ method which can be implemented using a Turing machine that runs in time $\mathrm{TIME}$.

A learning method $\mathcal{A}$ is said to be a *statistical-query* $\mathsf{SQ}(k, \tau, r)$ method, if it operates in $k$ rounds where in round $t$ the method produces a *statistical-query* $\Phi_t : \mathcal{X} \times \mathcal{Y} \to [-1, 1]$ for which it receives a response $v_t$, and finally outputs a predictor $f : \mathcal{X} \to \mathbb{R}$. Formally, each $\Phi_t$ is a function of a random string $R \in \{0, 1\}^r$ and past responses $v_1, \ldots, v_{t-1}$. We say that $\mathcal{A}$ ensures error $\varepsilon$ on a source distribution $\mathcal{D}$ if $\mathbb{E}_R[\sup \mathcal{L}_{\mathcal{D}}(f)] \leq \varepsilon$, where the sup is over all "valid" $v_t \in [-1, 1]$, namely $|v_t - \mathbb{E}_{\mathcal{D}} \Phi_t(x, y)| \leq \tau$. Similarly, we say that $\mathcal{A}$ is a $\mathsf{SQ}_{\mathsf{TM}}(k, \tau, r, \mathrm{TIME})$ algorithm if $\mathcal{A}$ is a $\mathsf{SQ}(k, \tau, r)$ method that can be implemented using a Turing machine that runs in time $\mathrm{TIME}$, using queries that can be computed in time $\mathrm{TIME}$.

**Relating Classes of Methods.**   The subject of this work is to compare between what can be learnt with different classes of methods. To do so, we define a general notion of when methods in class $\mathcal{C}$ can be *simulated by* methods in $\mathcal{C}'$, and thus $\mathcal{C}'$ can be said to be at least as powerful as $\mathcal{C}$. For any method/algorithm $\mathcal{A}$ and for any source distribution $\mathcal{D}$, let $\mathrm{err}(\mathcal{A}, \mathcal{D})$ be the infimum over $\varepsilon$ such that $\mathcal{A}$ ensures error $\varepsilon$ on $\mathcal{D}$. We now define:

**Definition 1.** *For two classes of methods $\mathcal{C}, \mathcal{C}'$, and any $\delta \geq 0$, we write $\mathcal{C}' \preceq_\delta \mathcal{C}$ if for every method $\mathcal{A} \in \mathcal{C}$ there exists a method $\mathcal{A}' \in \mathcal{C}'$ such that for every source distribution $\mathcal{D}$ we have $\mathrm{err}(\mathcal{A}', \mathcal{D}) \leq \mathrm{err}(\mathcal{A}, \mathcal{D}) + \delta$.*

That is, $\mathcal{C}' \preceq_\delta \mathcal{C}$ means that $\mathcal{C}'$ *is at least as powerful as* $\mathcal{C}$. Observe that for all classes of methods $\mathcal{C}_1, \mathcal{C}_2, \mathcal{C}_3$, if $\mathcal{C}_1 \preceq_{\delta_1} \mathcal{C}_2$ and $\mathcal{C}_2 \preceq_{\delta_2} \mathcal{C}_3$ then $\mathcal{C}_1 \preceq_{\delta_1 + \delta_2} \mathcal{C}_3$.

## 3   Main Results : bSGD versus PAC and SQ

Our main result, given below as a four-part Theorem, establishes the power of bSGD learning relative to PAC (i.e. arbitrary sample based) and SQ learning. As previously discussed, the exact relation depends on the mini-batch size $b$ and gradient precision $\rho$. First, we show that for any mini-batch size $b$, with fine enough precision $\rho$, bSGD can simulate PAC.

**Theorem 1a** (PAC to bSGD). *For all $b$ and $\rho < 1/(8b)$, and for all $m, r, \delta$, it holds that*

$$\mathsf{bSGD}(T' = O(mn/\delta), \boldsymbol{\rho}, \boldsymbol{b}, p' = r + O(T'(n + \log b)), r') \preceq_\delta \mathsf{PAC}(m, r).$$

*where $r' = r + O(T' \log b)$. Furthermore, if $\rho < \min\{1/(8b), 1/12\}$, using a piecewise linear activation function $\sigma$, for every runtime $\mathrm{TIME}$, it holds for $p' = \mathrm{poly}(n, m, r, \mathrm{TIME}, \rho^{-1}, b, \delta^{-1})$ and the same $T', r'$ above that*

$$\mathsf{bSGD}_{\mathsf{NN}}^\sigma(T', \boldsymbol{\rho}, \boldsymbol{b}, p', r') \preceq_\delta \mathsf{PAC}_{\mathsf{TM}}(m, r, \mathrm{TIME})$$

To establish equivalence (when Theorem 1a holds), we also note that PAC is always at least as powerful as bSGD (since bSGD can be implemented using samples):

**Theorem 1b** (bSGD to PAC). *For all $b$, $\rho$ and $T, p, r$, it holds that*

$$\mathsf{PAC}(m' = Tb, r' = r) \preceq_0 \mathsf{bSGD}(T, \boldsymbol{\rho}, \boldsymbol{b}, p, r).$$

*Furthermore, for all poly-time computable activations $\sigma$, it holds that*

$$\mathsf{PAC}_{\mathsf{TM}}(m' = Tb, r', \mathrm{TIME}' = \mathrm{poly}(T, b, p, r, n, \log \rho)) \preceq_0 \mathsf{bSGD}_{\mathsf{NN}}^\sigma(T, \boldsymbol{\rho}, \boldsymbol{b}, p, r).$$

If the mini-batch size is large relative to the precision, bSGD cannot go beyond SQ:

**Theorem 1c** (bSGD to SQ). *There exists a constant $C$ such that for all $\delta > 0$, for all $T$, $\rho$, $b$, $p$, $r$, such that $b\rho^2 > C \log(Tp/\delta)$, it holds that*

$$\mathsf{SQ}(k' = Tp, \tau' = \frac{\rho}{8}, r' = r) \preceq_\delta \mathsf{bSGD}(T, \boldsymbol{\rho}, \boldsymbol{b}, p, r).$$

*Furthermore, for all poly-time computable activations $\sigma$, it holds that*

$$\mathsf{SQ}_{\mathsf{TM}}\left(k' = Tp, \tau' = \frac{\rho}{8}, r' = r, \mathrm{TIME}' = \mathrm{poly}(T, \frac{1}{\rho}, b, p, r, \frac{1}{\delta})\right) \preceq_\delta \mathsf{bSGD}_{\mathsf{NN}}^\sigma(T, \boldsymbol{\rho}, \boldsymbol{b}, p, r).$$

To complete the picture, we also show that regardless of the mini-batch size, i.e. even when bSGD cannot simulate PAC, bSGD can always, at the very least, simulate any SQ method. This also establishes equivalence to SQ when Theorem 1c holds:

**Theorem 1d** (SQ to bSGD). *There exists a constant $C$ such that for all $\delta > 0$, **for all $b$** and all $k, \tau, r$, it holds that*

$$\mathsf{bSGD}(T', \boldsymbol{\rho} = \frac{\tau}{16}, \boldsymbol{b}, p' = r + 2T, r' = r) \ \preceq_\delta \ \mathsf{SQ}(k, \tau, r).$$

*where $T' = k \cdot \left\lceil \frac{C \log(k/\delta)}{b\tau^2} \right\rceil$. Furthermore, using a piecewise linear activation function $\sigma$, it holds for $p' = \mathrm{poly}(k, 1/\tau, r, \mathrm{TIME}, 1/\delta)$ that*

$$\mathsf{bSGD}^\sigma_{\mathsf{NN}}(T', \boldsymbol{\rho} = \frac{\tau}{16}, \boldsymbol{b}, p', r' = r) \ \preceq_\delta \ \mathsf{SQ}_{\mathsf{TM}}(k, \tau, r, \mathrm{TIME}).$$

In the above Theorems, the reductions hold with parameters on the left-hand-side (the parameters of the model being reduced to) that are polynomially related to the dimension and the parameters on the right-hand-side. But the mini-batch size $b$ and gradient precision $\rho$ play an important role. In Theorem 1a, we may choose $b$ and $\rho$ as we wish, as long as they satisfy $b\rho < 1/8$—they do not need to be chosen based on the parameters of the PAC method, and we can always simulate PAC with any $b$ and $\rho$ satisfying $b\rho < 1/8$. Similarly, in Theorem 1d, we may chose $b \geq 1$ arbitrarily, and can always simulate SQ, although $\rho$ does have to be chosen according to $\tau$. The reverse reduction of Theorem 1c, establishing the limit of when bSGD cannot go beyond SQ, is valid when $b\rho^2 = \omega(\log n)$, if the size of the model $p$ and number of SGD iterations $T$ are restricted to be polynomial in $n$.

Focusing on the mini-batch size $b$, and allowing all other parameters to be chosen to be polynomially related, Theorems 1a to 1d can be informally summarized as:

$$\mathsf{PAC} \ \preceq \ \mathsf{bSGD}(b) \ \preceq \ \mathsf{SQ} \tag{2}$$

where the left relationship is tight if $b < 1/(8\rho)$ and the right relationship is tight if $b > \omega((\log n)/\rho^2)$. Equivalently, focusing on the precision $\rho$ and how it depends on the mini-batch size $b$, and allowing all other parameters to be polynomially related, Theorems 1a to 1d can be informally summarized as:

$$\text{If } \rho < 1/(8b), \qquad\qquad \mathsf{bSGD}(b, \rho) \approx \mathsf{PAC} \tag{3}$$

$$\text{If } \rho > \omega\left(\sqrt{(\log n)/b}\right), \qquad \mathsf{PAC} \npreceq \mathsf{SQ}(\Theta(\rho)) \approx \mathsf{bSGD}(b, \rho) \tag{4}$$

$$\text{And in any case,} \qquad\qquad \mathsf{bSGD}(b, \rho) \preceq \mathsf{SQ}(\Theta(\rho)) \tag{5}$$

More formally, the results can also be viewed as a relationship between classes of learning problems. A *learning problem* is a sequence $(\mathcal{P}_n)_{n \in \mathbb{N}}$, where each $\mathcal{P}_n$ is a set of distributions $\mathcal{D}_n$ over $\{0, 1\}^n \times \mathcal{Y}$. For a parametrized class of methods $\mathcal{C}(\theta)$, we say a learning problem is poly-learnable with $\mathcal{C}$ if for every polynomial $\varepsilon(n)$ there is a polynomial $\theta(n)$ such that for every $n$ there is a method in $\mathcal{C}(\theta(n))$ that ensures error at most $\varepsilon(n)$ on all distributions in $\mathcal{P}_n$. We slightly abuse notation and use $\mathcal{C}$ to denote the set of learning problems poly-learnable with $\mathcal{C}$ methods, so that, e.g. PAC, $\mathsf{PAC}_{\mathsf{TM}}$, SQ and $\mathsf{SQ}_{\mathsf{TM}}$, are the familiar classes of PAC and SQ (poly) learnable problems. For bSGD we also define $\mathsf{bSGD}[b(n), \rho(n)]$, where $b$ and $\rho$ are only allowed to depend on $n$ via the specified (polynomially bounded, possibly constant) dependence, and not arbitrarily (the other parameters may grow as an arbitrary polynomial), and $\mathsf{bSGD}[b(n, \rho)]$, where $1/\rho$ can grow as an arbitrary polynomial, but the choice of $b$ as a function of $n$ and $\rho$ is constrained. With this notation, our results imply the following relationships between these classes of learning problems:

**Corollary 1.** *For any (poly bounded, possible constant) dependence $b(n, \rho)$, and for a piecewise linear activation function $\sigma$, it holds that*

$$\mathsf{SQ} \ \subseteq^{(1)} \ \mathsf{bSGD}[b] \qquad \subseteq^{(2)} \ \mathsf{PAC}$$

$$\mathsf{SQ}_{\mathsf{TM}} \ \subseteq^{(3)} \ \mathsf{bSGD}^\sigma_{\mathsf{NN}}[b] \qquad \subseteq^{(4)} \ \mathsf{PAC}_{\mathsf{TM}}$$

*Moreover, if $\forall_{n, \rho} \ b(n, \rho) < 1/(8\rho)$, then inclusions (2) and (4) are tight, and if $b(n, \rho) \geq \omega(\log n)/\rho^2$, then inclusions (1) and (3) are tight.*

**Corollary 2.** *For any (poly bounded, possibly constant) $b(n), \rho(n)$, and a piecewise linear activation function $\sigma$:*

$$\text{If } \forall_n \ b\rho < 1/8 \text{ then} \quad \mathsf{bSGD}[b, \rho] = \mathsf{PAC} \qquad \text{and} \quad \mathsf{bSGD}^\sigma_{\mathsf{NN}}[b, \rho] = \mathsf{PAC}_{\mathsf{TM}}$$

$$\text{If } b\rho^2 \geq \omega(\log n) \text{ then} \quad \mathsf{bSGD}[b, \rho] \subseteq \mathsf{SQ} \subsetneq \mathsf{PAC} \qquad \text{and} \quad \mathsf{bSGD}^\sigma_{\mathsf{NN}}[b, \rho] \subseteq \mathsf{SQ}_{\mathsf{TM}} \subsetneq \mathsf{PAC}_{\mathsf{TM}}$$

In Corollaries 1 and 2, for the sake of simplicity, we focused on *realizable* learning problems, where the minimal loss $\inf_f \mathcal{L}_{\mathcal{D}_n}(f) = 0$ for each $\mathcal{D}_n \in \mathcal{P}_n$. However, we note that Theorems 1a to 1d are more general, as they preserve the performance of learning methods (up to an additive $\delta$) on all source distributions $\mathcal{D}$. So, a result similar to Corollary 1 could be stated for other forms of learning, such as agnostic learning, weak learning etc.

**Proof Outline.** In order to prove Theorems 1a to 1d, we first relate bSGD to an intermediate model, mini-**b**atch **S**tatistical **Q**ueries (bSQ), which we introduce in Section 4. This is a variant of SQ-learning, but where statistical queries are answered based on mini-batches of samples. In the supplementary material we discuss how to simulate arbitrary statistical queries as gradient calculations for a specifically crafted model, and thus establish how to simulate (a large enough subclass of) bSQ methods using bSGD on some model $f_{\boldsymbol{w}}$. Furthermore, if the bSQ method has runtime TIME, then $f_{\boldsymbol{w}}$ is also computable in time $\mathrm{poly}(\text{TIME})$, and so can be implemented as a circuit, and thus a neural net of size $\mathrm{poly}(\text{TIME})$. With this ability to simulate bSQ methods with bSGD in mind, what remains is to relate bSQ to SQ and PAC, which is the subject of Section 4. Simulating an SQ method with a bSQ method is fairly straightforward, as each statistical query on the population can be simulated by averaging a large enough number of empirical statistical queries (with the resulting precision being bounded by the precision of each of the empirical statistical queries). This can be done using any sample size, unrelated to the precision. More surprising, we show how to simulate any sample based (PAC) method using bSQ, provided the precision is fine enough relative to the mini-batch size. This is done using a novel sample extraction procedure, which can extract a single sample sampled from $\mathcal{D}$ using a polynomial number of mini-batch statistical queries, and with precision linear in the mini-batch size (this is required so that each element in the mini-batch has a noticeable effect on the gradients). To complete the picture, we show that with low precision, statistical queries on a sample and on the population are indistinguishable (i.e. queries on an empirical mini-batch can be simulated by population queries, up to the required precision) and so neither bSQ nor bSGD, can go beyond SQ (establishing Theorem 1c). Finally, simulating bSGD using PAC (Theorem 1b) is straightforward, as bSGD is defined in terms of samples. The full proofs of Theorems 1a to 1d are presented in the supplementary material, with key components regarding bSQ developed in Section 4.

## 4 The Mini-Batch Statistical Query Model

En route to proving Theorems 1a to 1d, we introduce the model of mini-**b**atch **S**tatistical **Q**ueries (bSQ). In this model, similar to the standard Statistical Query (SQ) learning model, learning is performed through statistical queries. But in bSQ, these queries are answered based on an empirical average over a mini-batch of $b$ i.i.d. samples from the source distribution. That is, each query $\Phi_t : \mathcal{X} \times \mathcal{Y} \to [-1,1]^p$ is answered with a response $v_t$ s,t,

$$\left\| v_t - \frac{1}{b} \sum_{i=1}^{b} \Phi_t(x_{t,i}, y_{t,i}) \right\|_\infty \le \tau, \qquad \text{where } S_t = ((x_{t,i}, y_{t,i}))_i \sim \mathcal{D}^b \tag{6}$$

Note that we allow $p$-dimensional vector "queries", that is, $p$ concurrent scalar queries are answered based on the same mini-batch $S_t$, drawn independently for each vector query.

Formally, a learning method $\mathcal{A}$ is said to be a $\mathsf{bSQ}(k, \tau, b, p, r)$ method, if it operates in $k$ rounds where in round $t$, the method produces a query $\Phi_t : \mathcal{X} \times \mathcal{Y} \to [-1,1]^p$ for which it receives a response $v_t$ satisfying (6) and finally outputs a predictor $f : \mathcal{X} \to \mathbb{R}$. To be precise, each $\Phi_t$ is a function of a random string $R \in \{0,1\}^r$ and past responses $v_1, \ldots, v_{t-1}$, and the output is a function of the random string $R$ and all responses. We say that $\mathcal{A}$ ensures error $\varepsilon$ on a source distribution $\mathcal{D}$ if $\mathbb{E}[\sup \mathcal{L}_{\mathcal{D}}(f)] \le \varepsilon$, where the expectation is over $R$ and the mini-batches $S_t \sim \mathcal{D}^b$ and the sup is over all "valid" $v_t \in [-1,1]^p$ satisfying (6). A learning method is said to be $\mathsf{bSQ}_{\mathsf{TM}}(k, \tau, b, p, r, \text{TIME})$ if in addition it can be implemented with computational time at most TIME.

The first step of our simulation of PAC and SQ with bSGD is to simulate (a variant of) bSQ using bSGD. But beyond its use as an intermediate model in studying differentiable learning, bSQ can also be though of as a realistic way of answering statistical queries. In fact, one of the main justifications for allowing errors in the SQ model, and the demand that the error tolerance $\tau$ be polynomial, is that it is possible to answer statistical queries about the population with tolerance $\tau$ by calculating empirical averages on samples of size $O(1/\tau^2)$. In the bSQ model we make this explicit, and indeed answer

the queries using such samples. We do also allow additional arbitrary error beyond the sampling error, which we might think of as "precision". The bSQ model can thus be thought of as decomposing the SQ tolerance to a sampling error $O(1/\sqrt{b})$ and an additional arbitrary error $\tau$. If the arbitrary error $\tau$ indeed captures "precision", it is reasonable to take it to be exponentially small (corresponding to polynomially many bits of precision), while the sampling error would still be polynomial in a poly-time poly-sample method. Studying the bSQ model can reveal to us how well the standard SQ model captures what can be done when most of the error in answering statistical queries is due to the sampling error.

Our bSQ model is similar to the *honest-SQ* model studied by [20, 21], who also asked whether answering queries based on empirical averages changes the power of the model. But the two models have some significant differences, which lead to different conclusions, namely: Honest-SQ does not allow for an additional arbitrary error (i.e. it uses $\tau = 0$ in our notation), but an independent mini-batch is used for each single-bit query $\Phi_t : \mathcal{X} \times \mathcal{Y} \to \{0, 1\}$, whereas bSQ allows for $p$ concurrent real-valued scalar queries on the same mini-batch. Yang showed that, with a single bit query per mini-batch, and even if $\tau = 0$, it is not possible to simulate arbitrary sample-based methods, and honest-SQ is strictly weaker than PAC. But we show that once multiple bits can be queried concurrently[10] the situation is quite different.

In fact, we show that when the arbitrary error $\tau$ is small relative to the sample size $b$ (and thus the sampling error), bSQ can actually go well beyond SQ learning, and can in fact simulate any sample-based method. That is, SQ does not capture learning using statistical queries answered (to within reasonable precision) using empirical averages:

**Theorem 2a.** (PAC to bSQ) *For all $\delta > 0$, **for all $b$, and $\tau < 1/(2b)$**, and for all $m, r$, it holds for $k' = 10m(n+1)/\delta$, $p' = n + 1$, $r' = r + k \log_2 b$ that*
$$\mathsf{bSQ}(k', \boldsymbol{\tau}, \boldsymbol{b}, p', r') \preceq_\delta \mathsf{PAC}(m, r).$$

*Proof Sketch.* The main ingredient is a bSQ method SAMPLE-EXTRACT (Algorithm 1) that extracts a sample $(x, y) \sim \mathcal{D}$ by performing mini-batch statistical queries over independently sampled mini-batches. For ease of notation, let $\mathcal{Z} := \mathcal{X} \times \mathcal{Y}$ identifying it with $\{0, 1\}^{n+1}$ and denote $z \in \mathcal{Z}$ as $(z_1, \ldots, z_{n+1}) = (y, x_1, \ldots, x_n)$. SAMPLE-EXTRACT operates by sampling the bits of $z$ one by one, drawing $\widehat{z_i}$ from the conditional distribution $\{z_i \mid z_{1, \ldots, i-1}\}_\mathcal{D}$.

The main reason why $\tau < 1/2b$ enables this method to work is that for any query of the form $\Phi : \mathcal{X} \times \mathcal{Y} \to \{0, 1\}^p$, given any valid response $v$ such that $\|v - \mathbb{E}_S \Phi(x, y)\|_\infty \leq \tau$, it is possible to *exactly* recover $\mathbb{E}_S \Phi(x, y)$, by simply rounding each entry to the nearest integral multiple of $1/b$. We show that SAMPLE-EXTRACT returns a sample drawn from $\mathcal{D}$ in $O(n)$ expected number of queries. Thus, $k' = O(mn/\delta)$ queries suffice to recover $m$ samples, with a failure probability of at most $\delta$. Once we can simulate sample access, it is straightforward to simulate the entire PAC method. The details are deferred to the supplementary material. □

To complement the Theorem, we also note that sample-based learning is always at least as powerful as bSQ, since bSQ is specified based on a sample of size $kb$ (see the supplementary material for a complete proof):

**Theorem 2b.** (bSQ to PAC) *For all $b$, $\boldsymbol{\tau}$ and $k, p, r$, it holds that*
$$\mathsf{PAC}(m' = kb, r' = r) \preceq_0 \mathsf{bSQ}(k, \boldsymbol{\tau}, \boldsymbol{b}, p, r).$$

On the other hand, when the the mini-batch size $b$ is large relative to the precision (i.e. the arbitrary error $\tau$ is large relative to the sampling error $1/\sqrt{b}$), bSQ is no more powerful than standard SQ:

**Theorem 2c.** (bSQ to SQ) *There exists a constant $C \geq 0$ such that for all $\delta > 0$, **for all $k$, $\boldsymbol{\tau}$, $\boldsymbol{b}$, $p$, $r$**, such that $b\tau^2 > C \log(kp/\delta)$, it holds that*
$$\mathsf{SQ}(k' = kp, \tau' = \frac{\tau}{2}, r' = r) \preceq_\delta \mathsf{bSQ}(k, \boldsymbol{\tau}, \boldsymbol{b}, p, r).$$

---

[10]We do so with polynomially many binary-valued queries, i.e. $\Phi_t(x) \in \{0, 1\}^p$, and $p$ polynomial. It is also possible to encode this into a single real-valued query with polynomially many bits of precision. Once we use real-valued queries, if we do not limit the precision at all, i.e. $\tau = 0$, and do not worry about processing time, its easy to extract the entire minibatch $S_t$ using *exponentially* many bits of precision. Theorem 2a shows that *polynomially* many bits are sufficient for extracting a sample and simulating PAC.

---

**Algorithm 1** SAMPLE-EXTRACT bSQ algorithm (lines in **green** contain high-level idea of algorithm)

---

**Input:** Batch size $b$, Tolerance $\tau$ satisfying $b\tau < 1/2$.
**Output:** Sample $z \sim \mathcal{D}$

$s \leftarrow \epsilon$      **... (empty prefix string)**
**repeat**
   **# Let $S \sim \mathcal{D}^b$ be the independently sampled mini-batch**
   For $\ell := \text{length}(s)$, issue bSQ $\Phi : \mathcal{Z} \to \{0,1\}^{n-\ell+2}$, given as

$$\Phi_0(z) := \mathbb{1}\{z_{1,\ldots,\ell} = s\} \qquad\qquad \text{\# not required when } \ell = 0$$
$$\Phi_j(z) := \mathbb{1}\{z_{1,\ldots,\ell} = s \text{ and } z_{\ell+j} = 1\} \qquad \text{for all } 1 \le j \le n+1-\ell$$

   Let $v \in [0,1]^{n-\ell+2}$ be any valid answer, namely $\|v - \mathbb{E}_S \Phi(z)\|_\infty < \tau$ for independently
   sampled $S \sim \mathcal{D}^b$. Round each $v_i$ to the nearest integral multiple of $1/b$.
                                              # Since $\tau < 1/2b$, this ensures $v_i = \mathbb{E}_S[\Phi_i(z)]$.

$$\text{Let:} \quad w \leftarrow bv_0, \quad w_1 \leftarrow bv_1, \quad w_0 \leftarrow b(v_0 - v_1).$$

   **# $w$ / $w_0$ / $w_1 \leftarrow$ number of samples in $S$ that match prefix $s$ / $s \circ 0$ / $s \circ 1$.**
   **if** $w = 0$ **then**
      **# Do nothing; repeat the loop with a new sample.**       ▷ **(No sample matches prefix)**
   **else if** $w = 1$ **then**
      **# Return (unique) sample in $S$ that matches prefix $s$.**    ▷ **(Exactly one sample matches)**
      **return** $(s_1, \ldots, s_\ell, \widehat{z}_{\ell+1}, \ldots, \widehat{z}_{n+1})$, where $\widehat{z}_{\ell+j} = \mathbb{1}\{v_j = 1/b\}$ for all $1 \le j \le n+1-\ell$.
   **else**
      **# Extend prefix $s$ to reduce expected number of sample points with the prefix.**
$$s \leftarrow \begin{cases} s \circ 0 & \text{with probability } w_0/w \\ s \circ 1 & \text{with probability } w_1/w \end{cases} \qquad \text{▷ \textbf{(More than one sample matches prefix)}}$$
      **# Return $s$ if it fully specifies a sample point $(x, y)$.**
      **if** $\text{length}(s) = n+1$ : **return** $s$.
      **# Otherwise, repeat loop with the longer prefix $s$ (with new samples).**
   **end if**
**until** a sample is returned

---

*Proof Sketch.* When $b \gg 1/\tau^2$, the differences between the empirical and population averages become (with high probability) much smaller than the tolerance $\tau$, the population statistical query answers are valid responses to queries on the mini-batch, and we can thus simulate bSQ using SQ. We do need to make sure this holds uniformly for the $p$ parallel scalar queries, and across all $k$ rounds—see the supplementary material for a complete proof. $\qquad\square$

Finally, we show that with any mini-batch size, and enough rounds of querying, we can always simulate any SQ method using bSQ:

**Theorem 2d.** (SQ to bSQ) *There exists a constant $C \ge 0$ such that for all $\delta > 0$, **for all $b$ and all** $k, \tau, r$, it holds that*

$$\mathsf{bSQ}\left(k' = k \cdot \left\lceil \frac{C\log(k/\delta)}{b\tau^2} \right\rceil, \boldsymbol{\tau}' = \frac{\tau}{2}, \boldsymbol{b}, p' = 1, r' = r\right) \preceq_\delta \mathsf{SQ}(k, \tau, r).$$

*Proof Sketch.* To obtain an answer to a statistical query on the population, even if the sample-size $b$ per query is small, we can average the responses for the same query over multiple mini-batches (i.e. over multiple rounds). This allows us to reduce the sampling error arbitrarily, and leaves us with only the arbitrary error $\tau'$ (the arbitrary errors also get averaged, and since each element in the average is no larger than $\tau'$, the magnitude of this average is also no large than $\tau'$). See full proofs in the supplementary material. $\qquad\square$

Computationally tractable versions of Theorems 2a to 2d hold analogously relating $\mathsf{bSQ_{TM}}$ to $\mathsf{SQ_{TM}}/\mathsf{PAC_{TM}}$ with only a polynomial increase in runtime. We defer the formal statements to the supplementary material.

**Proof Sketch of Theorems 1a and 1d.**   In order to simulate PAC and SQ methods using bSGD, we first show how bSGD can simulate (a subclass of) bSQ (with corresponding mini-batch size and precision, and without any restriction on their relationship), and then rely in turn on Theorems 2a and 2d showing how bSQ can simulate PAC and SQ. We first show how a gradient computation can encode a statistical query, and use this, for a bSQ method $\mathcal{A}$, to construct a differentiable model $f$, that is defined in terms of the queries performed by $\mathcal{A}$ and their dependence on previous responses, such that bSGD on $f$ simulates $\mathcal{A}$. This construction does not rely on $\mathcal{A}$ being computationally tractable. We then note that if $\mathcal{A}$ is computable in time TIME, i.e. all the queries are computable in time TIME, and the mapping from responses to queries are likewise computable in time TIME, then these mappings can be implemented as circuits of size $\mathrm{poly}(\text{TIME})$, enabling us to implement $f$ as a neural network, with these circuits as subnetworks. Finally, we discuss how bSGD can be simulated on a neural network where all edges have trainable weights, using a modified version of the above construction, as well as a standalone alternate approach. Full details of these proofs are deferred to the supplementary material.

## 5  Summary and Discussion

We provided an almost tight characterization of the learning power of mini-batch SGD, relating it to the well-studied learning paradigms PAC and SQ, and thus (nearly) settling the question of "what can be learned using mini-batch SGD?". That single-sample SGD is able to simulate PAC learning was previously known, but we extended this result considerably, studied its limit, and showed that even outside this limit, bSGD can still always simulate SQ. A gap still remains, when the mini-batch size is between $1/\rho$ and $\log(n)/\rho^2$, where we do not know where bSGD sits between SQ and PAC. We furthermore showed that with sufficient (polynomial) precision, even full Gradient Descent on an empirical loss can simulate PAC learning.

It is tempting to view our results, which show the theoretical power of differentiable learning, as explaining the success of this paradigm. But we do not think that modern deep learning behaves similar to the constructions in our work. While we show how any SQ or PAC algorithm can be simulated, this requires a very carefully constructed network, with an extremely particular initialization, which doesn't look anything like deep learning in current practice. Our result certainly does *not* imply that SGD on a *particular* neural net can learn anything learnable by PAC or SQ, as this would imply that such network can learn any computationally tractable function,[11] which is known to be impossible (subject to mild cryptographic assumptions).

Rather, we view our work as guiding us as to what questions we should ask toward understanding how *actual* deep learning works. We see that understanding differentiable learning in such a broad generality as we did here is probably too strong, as it results in answers involving unrealistic initialization, and no restriction, and thus no insight, as to what makes learning problems learnable using deep learning. Can we define a class of neural networks, or initializations, which is broad enough to capture the power of deep learning, yet disallows such crazy initialization and does provide insight as to when deep learning is appropriate? Perhaps even mild restrictions on the initialization can already severely restrict the power of differentiable learning. E.g., Malach et al. [16] recently showed that even just requiring that the output of the network at initialization is close to zero can significantly change the power of differentiable learning, Abbe et al. [2] showed that imposing certain additional regularity assumptions on the architecture/initialization of neural networks restricts the learning power of (S)GD to function classes with a certain hierarchical property. An interesting direction for future work is understanding the power of differentiable learning under these, or other, restrictions. Does this lead to a different class of learnable problems, distinct from SQ and PAC, which is perhaps more related to deep learning in practice?

**Acknowledgements**

This work was done as part of the NSF-Simons Sponsored *Collaboration on the Theoretical Foundations of Deep Learning*. Part of this work was done while PK was at TTIC, and while NS was visiting EPFL. PK and NS were supported by NSF BIGDATA award 1546500 and NSF CCF/IIS award 1764032.

---

[11]Observe that for any tractable function $f$, there exists a trivial learning algorithm that returns $f$ regardless of its input, which means that the class $\{f\}$ is PAC learnable.

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
