is that of neural networks, defined by a directed acyclic graph with a single output node (i.e. with zero out-degree) and $n + 1$ input nodes (i.e. with zero

---

[3]Allowing $f_{\boldsymbol{w}}(x)$ to be non-differentiable on a measure zero set might seem overly generous, but it's simpler to state, and a more conservative restriction wouldn't affect our results. Our simulations use either everywhere differentiable models, or neural networks with piece-wise linear activation, resulting in a piece-wise linear models with a finite number of pieces.

in-degree) corresponding to the $n$ bits of $\mathcal{X}$ and the constant 1. Every edge $e$ corresponds to a weight parameter $w_e$. Computation proceeds recursively with each vertex $v$ returning the value $o_v$ obtained by applying an activation function $\sigma : \mathbb{R} \to \mathbb{R}$ on the linear combination of the values computed at its predecessors as specified by the edge weights $\boldsymbol{w} = (w_e)_e$, that is, $o_v := \sigma \left( \sum_{e=(u \to v)} w_e o_u \right)$. The final output of the model on input $x$ is the value returned by the output node.

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

. A method $\mathcal{A}$ is a fbGD$(T, \rho, m, p, r)$ if it operates similarly to a bSGD$(T, \rho, b = m, p, r)$ method with the only difference being that the same batch of samples is used at each iteration, namely $S_t = S$ for all $t$, where $S \sim \mathcal{D}^m$. Similarly, fbGD$^\sigma_{\text{NN}}(T, \rho, m, p, r)$ is defined analogous to bSGD$^\sigma_{\text{NN}}(T, \rho, b, p, r)$, where we require the differentiable model to be a neural network. As with bSGD methods, we say that $\mathcal{A}$ ensures error $\varepsilon$ on a source distribution $\mathcal{D}$ if $\mathbb{E}[\sup \mathcal{L}_D(f_{\boldsymbol{w}^{(T)}})] \leq \varepsilon$, where the expectation is over both the initialization $\boldsymbol{w}^{(0)}$ and the training set $S \sim \mathcal{D}^b$, and the $\sup$ is over all gradient estimates $g_t$ satisfying (3) at iterates where $f_{\boldsymbol{w}}$ is differentiable at $\boldsymbol{w}^{(t)}$ for all $x_i \in S$. Note that while bSGD$(T, \rho, b, p, r)$ uses $Tb$ samples overall, fbGD$(T, \rho, m, p, r)$ uses only $m$ samples in total.

**PAC and SQ Learning.**    A learning method $\mathcal{A}$ is said to be a PAC$(m, r)$ method[9] if it takes in a set of samples $S \subseteq \mathcal{X} \times \mathcal{Y}$ of size $m$, uses $r$ bits of randomness and returns a predictor $f : \mathcal{X} \to \mathbb{R}$. We say that $\mathcal{A}$ ensures error $\varepsilon$ on a source distribution $\mathcal{D}$ if $\mathbb{E}[\mathcal{L}_\mathcal{D}(f)] \leq \varepsilon$, where the expectation is over $S \sim \mathcal{D}^m$ and the randomness in $\mathcal{A}$. Similarly, we say that $\mathcal{A}$ is a PAC$_{\text{TM}}(m, r, \text{TIME})$ algorithm if it is a PAC$(m, r)$ method which can be implemented using a Turing machine that runs in time TIME.

A learning method $\mathcal{A}$ is said to be a *statistical-query* SQ$(k, \tau, r)$ method, if it operates in $k$ rounds where in round $t$ the method produces a *statistical-query* $\Phi_t : \mathcal{X} \times \mathcal{Y} \to [-1, 1]$ for which it receives a response $v_t$, and finally outputs a predictor $f : \mathcal{X} \to \mathbb{R}$. Formally, each $\Phi_t$ is a function of a random string $R \in \{0, 1\}^r$ and past responses $v_1, \ldots, v_{t-1}$. We say that $\mathcal{A}$ ensures error $\varepsilon$ on a source distribution $\mathcal{D}$ if $\mathbb{E}_R[\sup \mathcal{L}_\mathcal{D}(f)] \leq \varepsilon$, where the $\sup$ is over all "valid" $v_t \in [-1, 1]$, namely $|v_t - \mathbb{E}_\mathcal{D} \Phi_t(x, y)| \leq \tau$. Similarly, we say that $\mathcal{A}$ is a SQ$_{\text{TM}}(k, \tau, r, \text{TIME})$ algorithm if $\mathcal{A}$ is a SQ$(k, \tau, r)$ method that can be implemented using a Turing machine that runs in time TIME, using queries that can be computed in time TIME.

**Relating Classes of Methods.**    The subject of this work is to compare between what can be learnt with different classes of methods. To do so, we define a general notion of when methods in class $\mathcal{C}$ can be *simulated by* methods in $\mathcal{C}'$, and thus $\mathcal{C}'$ can be said to be at least as powerful as $\mathcal{C}$. For any method/algorithm $\mathcal{A}$ and for any source distribution $\mathcal{D}$, let $\text{err}(\mathcal{A}, \mathcal{D})$ be the infimum over $\varepsilon$ such that $\mathcal{A}$ ensures error $\varepsilon$ on $\mathcal{D}$. We now define:

**Definition 1.** *For two classes of methods $\mathcal{C}, \mathcal{C}'$, and any $\delta \geq 0$, we write $\mathcal{C}' \preceq_\delta \mathcal{C}$ if for every method $\mathcal{A} \in \mathcal{C}$ there exists a method $\mathcal{A}' \in \mathcal{C}'$ such that for every source distribution $\mathcal{D}$ we have $\text{err}(\mathcal{A}', \mathcal{D}) \leq \text{err}(\mathcal{A}, \mathcal{D}) + \delta$.*

That is, $\mathcal{C}' \preceq_\delta \mathcal{C}$ means that $\mathcal{C}'$ *is at least as powerful as* $\mathcal{C}$. Observe that for all classes of methods $\mathcal{C}_1, \mathcal{C}_2, \mathcal{C}_3$, if $\mathcal{C}_1 \preceq_{\delta_1} \mathcal{C}_2$ and $\mathcal{C}_2 \preceq_{\delta_2} \mathcal{C}_3$ then $\mathcal{C}_1 \preceq_{\delta_1 + \delta_2} \mathcal{C}_3$.

## 3    Main Results : bSGD versus PAC and SQ

Our main result, given below as a four-part Theorem, establishes the power of bSGD learning relative to PAC (i.e. arbitrary sample based) and SQ learning. As previously discussed, the exact relation depends on the mini-batch size $b$ and gradient precision $\rho$. First, we show that for any mini-batch size $b$, with fine enough precision $\rho$, bSGD can simulate PAC.

**Theorem 1a** (PAC to bSGD). *For all $b$ and $\rho < 1/(8b)$, and for all $m, r, \delta$, it holds that*

$$\text{bSGD}(T' = O(mn/\delta), \boldsymbol{\rho}, \boldsymbol{b}, p' = r + O(T'(n + \log b)), r') \preceq_\delta \text{PAC}(m, r).$$

$\mathbb{E}_{\boldsymbol{w}^{(0)}} \mathbb{E}_{S_1} \sup_{g_1} \ldots \mathbb{E}_{S_T} \sup_{g_T} \mathcal{L}_\mathcal{D}(f_{\boldsymbol{w}^{(T)}})$. However, this distinction does not change any of our results and we stick to the simpler $\mathbb{E} \sup$ definition for convenience.

[8]We say that $\sigma : \mathbb{R} \to \mathbb{R}$ is "polynomial-time computable" if there exists a Turing machine that, for any given $z \in \mathbb{R}$ and desired precision, computes $\sigma(z)$ and $\sigma'(z)$ to within the desired precision in time that is polynomial in the bit length of the representation of $z$ and the number of desired bits of precision.

[9]The usage of the acronym PAC here is technically improper as we use it to refer to a *method* and not a class of learning problems. We use it for notational convenience and historical reasons.

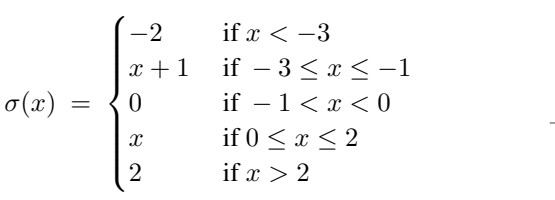
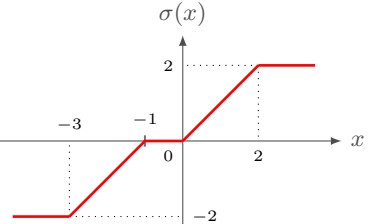

$$\sigma(x) \;=\; \begin{cases} -2 & \text{if } x < -3 \\ x+1 & \text{if } -3 \le x \le -1 \\ 0 & \text{if } -1 < x < 0 \\ x & \text{if } 0 \le x \le 2 \\ 2 & \text{if } x > 2 \end{cases}$$

Figure 1: Activation function used in Theorems 1a and 1d

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

$$\text{SQ} \subseteq^{(1)} \text{bSGD}[b] \qquad \subseteq^{(2)} \text{PAC}$$

$$\text{SQ}_{\text{TM}} \subseteq^{(3)} \text{bSGD}_{\text{NN}}^{\sigma}[b] \quad \subseteq^{(4)} \text{PAC}_{\text{TM}}$$

*Moreover, if $\forall_{n,\rho}\ b(n, \rho) < 1/(8\rho)$, then inclusions (2) and (4) are tight, and if $b(n, \rho) \geq \omega(\log n)/\rho^2$, then inclusions (1) and (3) are tight.*

**Corollary 2.** *For any (poly bounded, possibly constant) $b(n), \rho(n)$, and $\sigma$ from Figure 1:*

$$\text{If } \forall_n\ b\rho < 1/8 \text{ then } \quad \text{bSGD}[b, \rho] = \text{PAC} \qquad \text{and} \quad \text{bSGD}_{\text{NN}}^{\sigma}[b, \rho] = \text{PAC}_{\text{TM}}$$

$$\text{If } b\rho^2 \geq \omega(\log n) \text{ then } \quad \text{bSGD}[b, \rho] \subseteq \text{SQ} \subsetneq \text{PAC} \quad \text{and} \quad \text{bSGD}_{\text{NN}}^{\sigma}[b, \rho] \subseteq \text{SQ}_{\text{TM}} \subsetneq \text{PAC}_{\text{TM}}$$

In Corollaries 1 and 2, for the sake of simplicity, we focused on *realizable* learning problems, where the minimal loss $\inf_f \mathcal{L}_{\mathcal{D}_n}(f) = 0$ for each $\mathcal{D}_n \in \mathcal{P}_n$. However, we note that Theorems 1a to 1d are more general, as they preserve the performance of learning methods (up to an additive $\delta$) on all source distributions $\mathcal{D}$. So, a result similar to Corollary 1 could be stated for other forms of learning, such as agnostic learning, weak learning etc.

**Proof Outline.** In order to prove Theorems 1a to 1d, we first relate bSGD to an intermediate model, mini-**b**atch **S**tatistical **Q**ueries (bSQ), which we introduce in Section 4. This is a variant of SQ-learning, but where statistical queries are answered based on mini-batches of samples. In Section 5 we discuss how to simulate arbitrary statistical queries as gradient calculations for a specifically crafted model, and thus establish how to simulate (a large enough subclass of) bSQ methods using bSGD on some model $f_{\boldsymbol{w}}$. Furthermore, if the bSQ method has runtime TIME, then $f_{\boldsymbol{w}}$ is also computable in time $\text{poly}(\text{TIME})$, and so can be implemented as a circuit, and thus a neural net of size $\text{poly}(\text{TIME})$. With this ability to simulate bSQ methods with bSGD in mind, what remains is to relate bSQ to SQ and PAC, which is the subject of Section 4. Simulating an SQ method with a bSQ method is fairly straightforward, as each statistical query on the population can be simulated by averaging a large enough number of empirical statistical queries (with the resulting precision

being bounded by the precision of each of the empirical statistical queries). This can be done using any sample size, unrelated to the precision. More surprising, we show how to simulate any sample based (PAC) method using bSQ, provided the precision is fine enough relative to the mini-batch size. This is done using a novel sample extraction procedure, which can extract a single sample sampled from $\mathcal{D}$ using a polynomial number of mini-batch statistical queries, and with precision linear in the mini-batch size (this is required so that each element in the mini-batch has a noticeable effect on the gradients). To complete the picture, we show that with low precision, statistical queries on a sample and on the population are indistinguishable (i.e. queries on an empirical mini-batch can be simulated by population queries, up to the required precision) and so neither bSQ nor bSGD, can go beyond SQ (establishing Theorem 1c). Finally, simulating bSGD using PAC (Theorem 1b) is straightforward, as bSGD is defined in terms of samples. The full proofs of Theorems 1a to 1d are presented in Appendix D, with key ideas developed in Sections 4 and 5.

**Activation Functions and Fixed Weights.** The neural net simulations in Theorems 1a and 1d use a specific "stage-wise ramp" piecewise linear activation function with five linear pieces, depicted in Figure 1. A convenient property of this activation function is that it is has a central flat piece, making it easier for us to deal with weight drift due to rounding errors during training, and in particular drift of weights we would rather not change at all. Since any piecewise linear activation function can be simulated with ReLU activation, we could instead use a more familiar ReLU activation. However, the simulation "gadget" would involve weights that we would need fixed during training. That is, if we allow neural networks where some of the weights are fixed while others are trainable, we could use ReLU activation to simulate sample-based methods in Theorem 1a and SQ methods in Theorem 1d. As stated, we restrict ourselves only to neural nets where all edges have trainable weights, for which it is easier to prove the theorems with the specific activation function of Figure 1.

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

$$\mathsf{bSQ}_{\mathsf{TM}}(k', \boldsymbol{\tau}, \boldsymbol{b}, p', r', \mathrm{TIME}') \preceq_\delta \mathsf{PAC}_{\mathsf{TM}}(m, r, \mathrm{TIME}).$$

*Proof Sketch.* The main ingredient is a bSQ method SAMPLE-EXTRACT (Algorithm 1) that extracts a sample $(x, y) \sim \mathcal{D}$ by performing mini-batch statistical queries over independently sampled mini-batches. For ease of notation, let $\mathcal{Z} := \mathcal{X} \times \mathcal{Y}$ identifying it with $\{0,1\}^{n+1}$ and denote $z \in \mathcal{Z}$ as $(z_1, \ldots, z_{n+1}) = (y, x_1, \ldots, x_n)$. SAMPLE-EXTRACT operates by sampling the bits of $z$ one by one, drawing $\widehat{z}_i$ from the conditional distribution $\{z_i \mid z_{1,\ldots,i-1}\}_{\mathcal{D}}$.

The main reason why $\tau < 1/2b$ enables this method to work is that for any query of the form $\Phi : \mathcal{X} \times \mathcal{Y} \to \{0,1\}^p$, given any valid response $v$ such that $\|v - \mathbb{E}_S \Phi(x,y)\|_\infty \leq \tau$, it is possible to *exactly* recover $\mathbb{E}_S \Phi(x,y)$, by simply rounding each entry to the nearest integral multiple of $1/b$. In Appendix A.1, we show that SAMPLE-EXTRACT returns a sample drawn from $\mathcal{D}$ in $O(n)$ expected number of queries. Thus, $k' = O(mn/\delta)$ queries suffice to recover $m$ samples, with a failure probability of at most $\delta$. Once we can simulate sample access, it is straightforward to simulate the entire PAC method. The full proof is given in Appendix A.1. $\qquad\square$

To complement the Theorem, we also note that sample-based learning is always at least as powerful as bSQ, since bSQ is specified based on a sample of size $kb$ (see Appendix A.1 for a complete proof):

**Theorem 2b.** (bSQ to PAC) *For all $b$, $\tau$ and $k, p, r$, it holds that*

$$\mathsf{PAC}(m' = kb, r' = r) \preceq_0 \mathsf{bSQ}(k, \boldsymbol{\tau}, \boldsymbol{b}, p, r).$$

*Furthermore, for every runtime $\mathrm{TIME}$ it holds for $\mathrm{TIME}' = \mathrm{poly}(n, b, \mathrm{TIME})$ that*

$$\mathsf{PAC}_{\mathsf{TM}}(m' = kb, r' = r, \mathrm{TIME}') \preceq_0 \mathsf{bSQ}_{\mathsf{TM}}(k, \boldsymbol{\tau}, \boldsymbol{b}, p, r, \mathrm{TIME}).$$

On the other hand, when the the mini-batch size $b$ is large relative to the precision (i.e. the arbitrary error $\tau$ is large relative to the sampling error $1/\sqrt{b}$), bSQ is no more powerful than standard SQ:

**Theorem 2c.** (bSQ to SQ) *There exists a constant $C \geq 0$ such that for all $\delta > 0$, **for all $k$, $\tau$, $b$, $p$, $r$, such that $b\tau^2 > C \log(kp/\delta)$**, it holds that*

$$\mathsf{SQ}(k' = kp, \tau' = \frac{\tau}{2}, r' = r) \preceq_\delta \mathsf{bSQ}(k, \boldsymbol{\tau}, \boldsymbol{b}, p, r).$$

---

[10]We do so with polynomially many binary-valued queries, i.e. $\Phi_t(x) \in \{0,1\}^p$, and $p$ polynomial. It is also possible to encode this into a single real-valued query with polynomially many bits of precision. Once we use real-valued queries, if we do not limit the precision at all, i.e. $\tau = 0$, and do not worry about processing time, its easy to extract the entire minibatch $S_t$ using *exponentially* many bits of precision. Theorem 2a shows that *polynomially* many bits are sufficient for extracting a sample and simulating PAC.

---

**Algorithm 1** SAMPLE-EXTRACT bSQ algorithm (lines in **green** contain high-level idea of algorithm)

**Input:** Batch size $b$, Tolerance $\tau$ satisfying $b\tau < 1/2$.
**Output:** Sample $z \sim \mathcal{D}$

$s \leftarrow \epsilon$      **... (empty prefix string)**
**repeat**
   **# Let $S \sim \mathcal{D}^b$ be the independently sampled mini-batch**
   For $\ell := \text{length}(s)$, issue bSQ $\Phi : \mathcal{Z} \to \{0,1\}^{n-\ell+2}$, given as

$$\begin{aligned} \Phi_0(z) &:= \mathbb{1}\{z_{1,\ldots,\ell} = s\} && \text{\# not required when } \ell = 0 \\ \Phi_j(z) &:= \mathbb{1}\{z_{1,\ldots,\ell} = s \text{ and } z_{\ell+j} = 1\} && \text{for all } 1 \le j \le n+1-\ell \end{aligned}$$

   Let $v \in [0,1]^{n-\ell+2}$ be any valid answer, namely $\|v - \mathbb{E}_S\,\Phi(z)\|_\infty < \tau$ for independently
   sampled $S \sim \mathcal{D}^b$. Round each $v_i$ to the nearest integral multiple of $1/b$.
                                           # Since $\tau < 1/2b$, this ensures $v_i = \mathbb{E}_S[\Phi_i(z)]$.

            Let:     $w \leftarrow bv_0, \quad w_1 \leftarrow bv_1, \quad w_0 \leftarrow b(v_0 - v_1)$.

   **# $w$ / $w_0$ / $w_1 \leftarrow$ number of samples in $S$ that match prefix $s$ / $s \circ 0$ / $s \circ 1$.**
   **if** $w = 0$ **then**
     **# Do nothing; repeat the loop with a new sample.**        ▷ **(No sample matches prefix)**
   **else if** $w = 1$ **then**
     **# Return (unique) sample in $S$ that matches prefix $s$.**  ▷ **(Exactly one sample matches)**
     **return** $(s_1, \ldots, s_\ell, \widehat{z}_{\ell+1}, \ldots, \widehat{z}_{n+1})$, where $\widehat{z}_{\ell+j} = \mathbb{1}\{v_j = 1/b\}$ for all $1 \le j \le n+1-\ell$.
   **else**
     **# Extend prefix $s$ to reduce expected number of sample points with the prefix.**
     $s \leftarrow \begin{cases} s \circ 0 & \text{with probability } w_0/w \\ s \circ 1 & \text{with probability } w_1/w \end{cases}$     ▷ **(More than one sample matches prefix)**
     **# Return $s$ if it fully specifies a sample point $(x, y)$.**
     **if** $\text{length}(s) = n + 1$ : **return** $s$.
     **# Otherwise, repeat loop with the longer prefix $s$ (with new samples).**
   **end if**
**until** a sample is returned

---

*Furthermore, for any runtime* TIME *it holds for* $\text{TIME}' = \text{poly}(\text{TIME})$ *that*

$$\mathsf{SQ}_{\mathsf{TM}}\left(k' = kp, \tau' = \frac{\tau}{2}, r' = r, \text{TIME}'\right) \preceq_\delta \mathsf{bSQ}_{\mathsf{TM}}(k, \boldsymbol{\tau}, \boldsymbol{b}, p, r, \text{TIME}).$$

*Proof Sketch.* When $b \gg 1/\tau^2$, the differences between the empirical and population averages become (with high probability) much smaller than the tolerance $\tau$, the population statistical query answers are valid responses to queries on the mini-batch, and we can thus simulate bSQ using SQ. We do need to make sure this holds uniformly for the $p$ parallel scalar queries, and across all $k$ rounds—see [Appendix A.3](#) for a complete proof. □

Finally, we show that with any mini-batch size, and enough rounds of querying, we can always simulate any SQ method using bSQ:

**Theorem 2d.** (SQ to bSQ) *There exists a constant $C \ge 0$ such that for all $\delta > 0$, **for all $b$** and all $k, \tau, r$, it holds that*

$$\mathsf{bSQ}\left(k' = k \cdot \left\lceil \frac{C \log(k/\delta)}{b\tau^2} \right\rceil, \boldsymbol{\tau}' = \frac{\tau}{2}, \boldsymbol{b}, p' = 1, r' = r\right) \preceq_\delta \mathsf{SQ}(k, \tau, r).$$

*Furthermore, for every runtime* TIME *it holds for* $\text{TIME}' = \text{poly}(\text{TIME}, k, 1/\tau)$ *and the same* $k', \tau', p', r'$, *that*

$$\mathsf{bSQ}_{\mathsf{TM}}\left(k', \boldsymbol{\tau}', \boldsymbol{b}, p', r', \text{TIME}'\right) \preceq_\delta \mathsf{SQ}_{\mathsf{TM}}(k, \tau, r, \text{TIME}).$$

*Proof Sketch.* To obtain an answer to a statistical query on the population, even if the sample-size $b$ per query is small, we can average the responses for the same query over multiple mini-batches

(i.e. over multiple rounds). This allows us to reduce the sampling error arbitrarily, and leaves us with only the arbitrary error $\tau'$ (the arbitrary errors also get averaged, and since each element in the average is no larger than $\tau'$, the magnitude of this average is also no large than $\tau'$). See full proofs in Appendix A.3. □

## 5    Simulating Mini-Batch Statistical Queries with Differentiable Learning

As promised in Section 3, in order to simulate PAC and SQ methods using bSGD, and thus establish Theorems 1a and 1d, we first show how bSGD can simulate (a subclass of) bSQ (with corresponding mini-batch size and precision, and without any restriction on their relationship), and then rely in turn on Theorems 2a and 2d showing how bSQ can simulate PAC and SQ. We first show how a gradient computation can encode a statistical query, and use this, for a bSQ method $\mathcal{A}$, to construct a differentiable model $f$, that is defined in terms of the queries performed by $\mathcal{A}$ and their dependence on previous responses, such that bSGD on $f$ simulates $\mathcal{A}$. This construction does not rely on $\mathcal{A}$ being computationally tractable. We then note that if $\mathcal{A}$ is computable in time TIME, i.e. all the queries are computable in time TIME, and the mapping from responses to queries are likewise computable in time TIME, then these mappings can be implemented as circuits of size $\mathrm{poly}(\mathrm{TIME})$, enabling us to implement $f$ as a neural network, with these circuits as subnetworks.

**Alternating Query Methods.**    Instead of working with, and simulating, any bSQ method, we consider only *alternating* methods, denoted $\mathrm{bSQ}^{0/1}$, where in each round, only one of the two possible labels is involved in the query. Formally, we say that a (mini-batch) statistical query $\Phi : \mathcal{X} \times \mathcal{Y} \to [-1, 1]^p$ is a $\overline{y}$-query for $\overline{y} \in \mathcal{Y}$ if $\Phi(x, y) = 0$ for all $y \neq \overline{y}$, or equivalently $\Phi(x, y) = \mathbb{1}_{\{y = \overline{y}\}} \cdot \Phi_{\mathcal{X}}(x)$ for some $\Phi_{\mathcal{X}} : \mathcal{X} \to [-1, 1]^p$. A $\mathrm{bSQ}^{0/1}$ (analogously, $\mathrm{bSQ}^{0/1}_{\mathrm{TM}}$) method is a bSQ (analogously, $\mathrm{bSQ}_{\mathrm{TM}}$) method such that for all odd rounds $t$, $\Phi_t$ is a 1-query, and at all even rounds $t$, $\Phi_t$ is a 0-query. As minor extensions of Theorems 2a and 2d (simulation PAC and SQ using bSQ methods), we show that these simulations can in-fact be done using alternating queries, thus relating PAC and SQ to $\mathrm{bSQ}^{0/1}$. We present the full details in Appendices A.1 and A.3 respectively.

**Lemma 1.**  (PAC to $\mathrm{bSQ}^{0/1}$) *For all $\delta > 0$, **for all $b$, and $\tau < 1/(2b)$**, then and for all $m, r$, it holds for $k' = 20m(n+1)/\delta$, $p' = n + 1$, $r' = r + k \log_2 b$ that*

$$\mathrm{bSQ}^{0/1}(k', \boldsymbol{\tau}, \boldsymbol{b}, p', r') \preceq_{\delta} \mathrm{PAC}(m, r) .$$

*Furthermore, for every runtime TIME it holds for $\mathrm{TIME}' = \mathrm{poly}(n, m, r, \mathrm{TIME}, 1/\delta)$ that*

$$\mathrm{bSQ}^{0/1}_{\mathrm{TM}}(k', \boldsymbol{\tau}, \boldsymbol{b}, p', r', \mathrm{TIME}') \preceq_{\delta} \mathrm{PAC}_{\mathrm{TM}}(m, r, \mathrm{TIME}) .$$

**Lemma 2.**  (SQ to $\mathrm{bSQ}^{0/1}$) *There exists a constant $C > 0$, such that for all $\delta > 0$, **for all $b$** and all $k, \tau, r$, it holds that*

$$\mathrm{bSQ}^{0/1}\left(k' = k \cdot \left\lceil \frac{C \log(k/\delta)}{b\tau^2} \right\rceil, \boldsymbol{\tau}' = \frac{\tau}{4}, \boldsymbol{b}, p' = 1, r' = r\right) \preceq_{\delta} \mathrm{SQ}(k, \tau, r) .$$

*Furthermore, for every runtime TIME. it holds for $\mathrm{TIME}' = \mathrm{poly}(\mathrm{TIME}, k, 1/\tau)$ and the same $k'$, $\boldsymbol{\tau}'$, $p', r'$, that*

$$\mathrm{bSQ}^{0/1}_{\mathrm{TM}}(k', \boldsymbol{\tau}', \boldsymbol{b}, p', r', \mathrm{TIME}') \preceq_{\delta} \mathrm{SQ}_{\mathrm{TM}}(k, \tau, r, \mathrm{TIME}) .$$

**Simulating $\mathrm{bSQ}^{0/1}$ with differentiable programming.**    We now show how to to simulate a $\mathrm{bSQ}^{0/1}$ method with bSGD, with corresponding mini-batch and precision:

**Lemma 3a.**  ($\mathrm{bSQ}^{0/1}$ to bSGD) *For all $\tau \in (0, 1)$ and $b, k, p, r \in \mathbb{N}$, it holds that*

$$\mathrm{bSGD}\left(T' = k, \rho = \frac{\tau}{4}, b, p' = r + (p+1)k, r' = r\right) \preceq_0 \mathrm{bSQ}^{0/1}(k, \tau, b, p, r) .$$

*Proof Sketch.*  We first show how a single $\overline{y}$-query can be simulated using a single step of bSGD on a specific differentiable model. Given a 0-query $\Phi : \mathcal{X} \times \mathcal{Y} \to [-1, 1]^p$, consider the following model:

$$f_{\boldsymbol{w}}(x) = 1 - \langle \Phi_{\mathcal{X}}(x), \boldsymbol{w} \rangle \tag{9}$$

With $\boldsymbol{w}^{(0)} = 0$, the model $f_{\boldsymbol{w}^{(0)}}$ "guesses" the label to be $1$ for all examples, and therefore suffers a loss only for examples with the label $0$. Using a simple gradient calculations we get:

$$\nabla_{\boldsymbol{w}} \mathbb{E}_S \, \ell_{\text{sq}}(f_{\boldsymbol{w}^{(0)}}(x), y) \; = \; \mathbb{E}_S \, \mathbb{1}\{y = 0\} \cdot \nabla_{\boldsymbol{w}} f_{\boldsymbol{w}^{(0)}}(x) \; = \; -\mathbb{E}_S \, \mathbb{1}\{y = 0\} \cdot \Phi_{\mathcal{X}}(x) \; = \; -\mathbb{E}_S \Phi(x, y)$$

Hence, after a single step of bSGD we have $\boldsymbol{w}^{(1)} = \mathbb{E}_S \Phi(x, y)$ (up to precision $\rho$), so $\boldsymbol{w}^{(1)}$ stores the answer for the $0$-query $\Phi$. We can analogously simulate a $1$-query by setting the output to be $0$ at $\boldsymbol{w}^{(0)}$. We achieve the simulation of the complete bSQ$^{0/1}$ method using a composition of such differentiable models: for each round $t$ we reserve a parameter $\boldsymbol{w}_t$. The model is defined so that based on how many queries $t$ were already answered, and the responses as encoded in $\boldsymbol{w}_0, \dots, \boldsymbol{w}_{t-1}$, the objective is set to be locally linear in $\boldsymbol{w}_t$, with coefficients corresponding to the desired query, as above. These coefficients are of course a function of the other parameters. But the key is that the dependence is piecewise linear, and the dependence on all parameters other than $\boldsymbol{w}_t$ is constant around $\boldsymbol{w}$, ensuring that the only non-zero derivative is w.r.t. $\boldsymbol{w}_t$. Some complications needed to be overcome include: keeping track of how many queries were already executed (i.e. a "clock"), errors in gradients, and defining the model so that it is differentiable always everywhere and yet piecewise linear with the correct coefficients at points actually reached during training. Full details of the simulation and its proof are given in Appendix B. $\qquad\square$

Combining Lemma 3a with Lemmas 1 and 2 establishes the first statements (about computationally *unbounded* learning) of Theorems 1a and 1d; full details in Appendix D.

**Implementing the differentiable model as a Neural Network.** The simulation in Lemma 3a, as described above, uses some arbitrary differentiable model $f_{\boldsymbol{w}}(x)$, that is defined in terms of the mappings from responses to queries in the bSQ$^{0/1}$ method. If the bSQ$^{0/1}$ method is computationally bounded, the simulation can also be done using a neural network:

**Lemma 3b.** (bSQ$^{0/1}_{\text{TM}}$ to bSGD$^{\sigma}_{\text{NN}}$) *For all $\tau \in (0, \frac{1}{3})$ and $b, k, p, r \in \mathbb{N}$, and using the activation $\sigma$ from Figure 1, it holds that*

$$\text{bSGD}^{\sigma}_{\text{NN}} \left( T' = k, \rho = \frac{\tau}{4}, b, p' = \text{poly}(k, \frac{1}{\tau}, b, p, r, \text{TIME}), r' = r \right) \; \preceq_0 \; \text{bSQ}^{0/1}_{\text{TM}}(k, \tau, b, p, r, \text{TIME}) \, .$$

*Proof Sketch.* For a time-bounded bSQ$^{0/1}_{\text{TM}}$ method, the mapping from previous responses to the next query is computable in time TIME, and hence with a circuit, or neural network (with any non-trivial activation), of size $\text{poly}(\text{TIME})$. We can thus replace this mapping with a subnet computing it, and obtain a neural network implementing the differentiable model from Lemma 3a. This is a simple approach for obtaining a neural network where some of the weights are fixed (the weights in the subnetworks used to implement the mappings, as well as the "gating" between them) and only some of the edges have trainable weights. We are interested in simulating using bSGD on a neural network where all edges have trainable weights. In Appendix B.1 we discuss how the construction could be modified so that the edge weights in these subnetworks remain fixed over training, but with some compromises. Instead, in Appendix C we describe an alternate construction, following the same ideas as described in the proof sketch of Lemma 3a, but directly constructed as a neural network: we use the subnets implementing the mapping from responses to queries discussed above, where the initialization is very specific and encodes these functions, but design them in such a way that the vertices are always at flat parts of their activation functions so that it does not change. Then, for each query we designate one edge whose weight is intended to encode the result of that query, and connect the output of that edge to the net's output by means of a path with some vertices that the computation component can force to flat parts of their activation functions. That allows it to make the weights of those edges change in the desired manner by controlling the derivative of the net's output with respect to these weights. The full details are in Appendix C. $\qquad\square$

Combining Lemma 3b with Lemmas 1 and 2 establishes the second statements (about computationally *bounded* learning) of Theorems 1a and 1d; full details in Appendix D.

**Reverse direction: Simulating bSGD with PAC and SQ.** In order to establish Theorems 1b and 1c we rely on Theorems 2b and 2c and for that purpose note that bSGD can be directly implemented using bSQ (proof in Appendix D):

**Lemma 4.** (bSGD to bSQ) *For all $T, \rho, b, p, r$, it holds that*

$$\mathsf{bSQ}\left(k = T, \tau = \frac{\rho}{4}, b, p, r\right) \preceq_0 \mathsf{bSGD}(T, \rho, b, p, r).$$

*Furthermore, for every poly-time computable activation $\sigma$, it holds that*

$$\mathsf{bSQ_{TM}}\left(k = T, \tau = \frac{\rho}{4}, b, p, r, \mathrm{TIME} = \mathrm{poly}(T, p, b, r)\right) \preceq_0 \mathsf{bSGD_{NN}^\sigma}(T, \rho, b, p, r).$$

## 6 Full-Batch Gradient Descent: fbGD versus PAC and SQ

So far we considered learning with mini-batch stochastic gradient descent (bSGD), where an independent mini-batch of examples is used at each step. But this stochasticity, and the use of independent fresh samples for each gradient step, is not crucial for simulating PAC and SQ, provided enough samples overall, and a correspondingly fine enough precision. We show that analogous results hold for learning with *full-batch Gradient Descent* (fbGD), i.e. gradient descent on the (fixed) empirical loss.

**Theorem 3a** (PAC to fbGD). ***For all $m$ and $\rho < 1/(8m)$ and for all $r$, it holds that***

$$\mathsf{fbGD}(T' = O(mn), \boldsymbol{\rho}, \boldsymbol{m'} = m, p', r') \preceq_0 \mathsf{PAC}(m, r).$$

*where $p' = r + O(mn)$ and $r' = r$. Furthermore, using the piece-wise linear "two-stage ramp" activation $\sigma$ (Figure 1), for every runtime $\mathrm{TIME}$, it holds for $p' = \mathrm{poly}(n, m, r, \mathrm{TIME}, \rho^{-1})$ and the same $T', r'$ that*

$$\mathsf{fbGD_{NN}^\sigma}(T', \boldsymbol{\rho}, \boldsymbol{m'} = m, p', r') \preceq_0 \mathsf{PAC_{TM}}(m, r, \mathrm{TIME})$$

**Theorem 3b** (fbGD to PAC). ***For all $m$, $\boldsymbol{\rho}$ and $T, p, r$, it holds that***

$$\mathsf{PAC}(m' = \boldsymbol{m}, r' = r) \preceq_0 \mathsf{fbGD}(T, \boldsymbol{\rho}, \boldsymbol{m}, p, r).$$

*Furthermore, for all poly-time computable activations $\sigma$, it holds that*

$$\mathsf{PAC_{TM}}(m' = \boldsymbol{m}, r' = r, \mathrm{TIME'} = \mathrm{poly}(T, m, p, r, n)) \preceq_0 \mathsf{fbGD_{NN}^\sigma}(T, \boldsymbol{\rho}, \boldsymbol{m}, p, r, s).$$

**Theorem 3c** (fbGD to SQ). *There exists a constant $C$ such that for all $\delta > 0$, **for all $T$, $\boldsymbol{\rho}$, $\boldsymbol{m}$, $p$, $r$**, such that $\boldsymbol{m\rho^2} > C(\boldsymbol{Tp} \log(1/\rho) + \log(1/\boldsymbol{\delta}))$, it holds that*

$$\mathsf{SQ}(k' = Tp, \tau' = \frac{\rho}{8}, r' = r) \preceq_\delta \mathsf{fbGD}(T, \boldsymbol{\rho}, \boldsymbol{m}, p, r).$$

*Furthermore, for all poly-time computable activations $\sigma$, it holds that*

$$\mathsf{SQ_{TM}}\left(k' = Tp, \tau' = \frac{\rho}{8}, r' = r, \mathrm{TIME'} = \mathrm{poly}(T, \rho^{-1}, m, p, r, \delta^{-1})\right) \preceq_\delta \mathsf{fbGD_{NN}^\sigma}(T, \boldsymbol{\rho}, \boldsymbol{m}, p, r).$$

**Theorem 3d** (SQ to fbGD). *There exists a constant $C$ such that for all $\delta > 0$, for all $k, \tau, r$, it holds **for $m, \rho$ such that $\rho = \tau/16$, $m\rho^2 > C(k \log(1/\rho) + \log(1/\delta))$** that*

$$\mathsf{fbGD}(T' = 2k, \boldsymbol{\rho}, \boldsymbol{m}, p' = r + 2T, r' = r) \preceq_\delta \mathsf{SQ}(k, \tau, r).$$

*Furthermore, using the piece-wise linear "two-stage ramp" activation $\sigma$ (Figure 1), it holds for the same $T', r'$ above and $p' = \mathrm{poly}(k, 1/\tau, r, \mathrm{TIME}, 1/\delta)$ that*

$$\mathsf{fbGD_{NN}^\sigma}(T', \boldsymbol{\rho}, \boldsymbol{m}, p', r') \preceq_\delta \mathsf{SQ_{TM}}(k, \tau, r, \mathrm{TIME}).$$

The above theorems are analogous to Theorems 1a to 1d. They are proved in an analogous manner in Appendix E, by going through the intermediate model of *fixed-batch statistical query* fbSQ, in place of bSQ. An fbSQ$(k, \tau, m, p, r)$ method is described identically to an bSQ$(k, \tau, b = m, p, r)$ method, except that the responses for all queries are obtained using the same batch of samples in all rounds (i.e. $S_t = S$ for all $t$, where $S \sim \mathcal{D}^m$ in Equation (8)). Simulating fbSQ methods using fbGD, or fbGD$_{\mathsf{NN}}$ can be done using the exact same constructions as in Lemmas 3a and 3b. To establish Theorem 3a, we use an algorithm similar to (and simpler than) Algorithm 1 to extract all the samples batch of samples (see details in Lemma 5a in Appendix A.2). Relating fbSQ to SQ and establishing Theorems 3c and 3d requires more care, because of the adaptive nature of fbGD on the full-batch. Instead, we consider all possible queries the method might make, based on previous responses. Since

we have at most $Tp\log(1/\rho)$ or $kp\log(1/\tau)$ bits of response to choose a new query based on, we need to take a union bound over a number of queries exponential in this quantity, which results in the sample sized required to ensure validity scaling linear in $kp\log(1/\tau)$. See Appendix A.4 for complete proofs and details.

Theorem 3a tells us that with fine enough precision, even fbGD can simulate any sample-based learning method, and is thus as powerful as PAC. The precision required is linear in the total number of samples $m$ used by the method, i.e. the number of bits of precision is logarithmic in $m$. In particular, this implies that $\rho = \text{poly}(n)$, i.e $O(\log n)$ bits of precision, are sufficient for simulating any sample-based method that uses polynomially many samples. Returning to the relationship between classes of learning problems considered in Corollaries 1 and 2, where the parameters are allowed to depend polynomially on $n$, we have:

**Corollary 3.** $\text{fbGD} = \text{PAC}$ *and* $\text{fbGD}_{\text{NN}} = \text{PAC}_{\text{TM}}$.

But a significant difference versus bSGD is that with fbGD the precision depends (even if only polynomially) on the total number of samples used by the method. This is in contrast to bSGD, where the precision only has to be related to the mini-batch size used, and with constant precision (and constant mini-batch size), we could simulate any sample based based method, regardless of the number of samples used by the method (only the number $T$ of SGD iterations and the size $p$ of the model increase with the number of samples used). Viewed differently, consider what can be done with some fixed precision $\rho$ (that is not allowed to depend on the problem size $n$ or sample size $m$): methods that use up to $1/(8\rho)$ samples can be simulated even with fbGD. But bSGD allows us to simulate methods that use even more samples, by keeping the mini-batch size below $1/(8\rho)$.

It should also be noted that the limit of what can be done with fbGD, and when it cannot go beyond SQ, is not as clear and tight as for bSGD. Theorem 3c tells us that once $m = \tilde{\Omega}(Tp/\rho^2)$, we cannot go beyond SQ. But this bound on the sample size depends *polynomially* on the fbGD model size $p$ and number of iterations $T$ (as opposed to the logarithmic dependence in Theorem 1c). Even if the precision $\rho$ is bounded, it is conceivably possible to go beyond SQ and simulate any sample based method by using a polynomially larger model size $p$ and/or number of iterations iterations $T$ (and in any case $T$ and $p$ need to increase polynomially with $m$ when using the simulations of Lemmas 3a and 3b, even if using bSGD). It thus remains open whether it is possible to simulate any sample based method with fbGD using constant precision $\rho$ and where the model size $p$ and number of iterations $T$ are polynomial in the sample size $m$ and dimension $n$.

**Learning with mini-batch stochastic gradients over a fixed training set.** Perhaps the most realistic differentiable learning approach is to use a fixed training set $S$, and then at each iteration calculate a gradient estimate based on a mini-batch $S_t \subset S$ chosen at random, with replacement, from within the training set $S$ (as opposed to using fresh samples from the population distribution, as in bSGD). Analogs of Theorems 1a and 1d and Theorem 3c should hold also for this hybrid class, but we do not provide details here.

## 7 Summary and Discussion

We provided an almost tight characterization of the learning power of mini-batch SGD, relating it to the well-studied learning paradigms PAC and SQ, and thus (nearly) settling the question of "what can be learned using mini-batch SGD?". That single-sample SGD is able to simulate PAC learning was previously known, but we extended this result considerably, studied its limit, and showed that even outside this limit, bSGD can still always simulate SQ. A gap still remains, when the mini-batch size is between $1/\rho$ and $\log(n)/\rho^2$, where we do not know where bSGD sits between SQ and PAC. We furthermore showed that with sufficient (polynomial) precision, even full Gradient Descent on an empirical loss can simulate PAC learning.

It is tempting to view our results, which show the theoretical power of differentiable learning, as explaining the success of this paradigm. But we do not think that modern deep learning behaves similar to the constructions in our work. While we show how any SQ or PAC algorithm can be simulated, this requires a very carefully constructed network, with an extremely particular initialization, which doesn't look anything like deep learning in current practice. Our result certainly does *not* imply that SGD on a *particular* neural net can learn anything learnable by PAC or SQ, as this would imply that

such network can learn any computationally tractable function,[11] which is known to be impossible (subject to mild cryptographic assumptions).

Rather, we view our work as guiding us as to what questions we should ask toward understanding how *actual* deep learning works. We see that understanding differentiable learning in such a broad generality as we did here is probably too strong, as it results in answers involving unrealistic initialization, and no restriction, and thus no insight, as to what makes learning problems learnable using deep learning. Can we define a class of neural networks, or initializations, which is broad enough to capture the power of deep learning, yet disallows such crazy initialization and does provide insight as to when deep learning is appropriate? Perhaps even mild restrictions on the initialization can already severely restrict the power of differentiable learning. E.g., Malach et al. [16] recently showed that even just requiring that the output of the network at initialization is close to zero can significantly change the power of differentiable learning, Abbe et al. [2] showed that imposing certain additional regularity assumptions on the architecture/initialization of neural networks restricts the learning power of (S)GD to function classes with a certain hierarchical property. An interesting direction for future work is understanding the power of differentiable learning under these, or other, restrictions. Does this lead to a different class of learnable problems, distinct from SQ and PAC, which is perhaps more related to deep learning in practice?

## Acknowledgements

This work was done as part of the NSF-Simons Sponsored *Collaboration on the Theoretical Foundations of Deep Learning*. Part of this work was done while PK was at TTIC, and while NS was visiting EPFL. PK and NS were supported by NSF BIGDATA award 1546500 and NSF CCF/IIS award 1764032.

---

[11]Observe that for any tractable function $f$, there exists a trivial learning algorithm that returns $f$ regardless of its input, which means that the class $\{f\}$ is PAC learnable.

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

# A  Reductions between bSQ, fbSQ and SQ, PAC

In this section, we prove Theorems 2a to 2d. Additionally, we state and prove analogous statements relating fbSQ to PAC and SQ.

## A.1  bSQ versus PAC

**Theorem 2a.** (PAC to bSQ) *For all $\delta > 0$, **for all $b$, and $\tau < 1/(2b)$**, and for all $m, r$, it holds for $k' = 10m(n+1)/\delta$, $p' = n+1$, $r' = r + k\log_2 b$ that*

$$\mathsf{bSQ}(k', \boldsymbol{\tau}, \boldsymbol{b}, p', r') \preceq_\delta \mathsf{PAC}(m, r)\,.$$

*Furthermore, for every runtime $\mathrm{TIME}$, it holds for $\mathrm{TIME}' = \mathrm{poly}(n, m, r, \mathrm{TIME}, 1/\delta)$ that*

$$\mathsf{bSQ}_{\mathsf{TM}}(k', \boldsymbol{\tau}, \boldsymbol{b}, p', r', \mathrm{TIME}') \preceq_\delta \mathsf{PAC}_{\mathsf{TM}}(m, r, \mathrm{TIME})\,.$$

*Proof.* For all $b, \tau$ satisfying $b\tau < 1/2$, we first design a $\mathsf{bSQ}(k = 10(n+1), \tau, b, p = n+1, r)$ algorithm SAMPLE-EXTRACT (Algorithm 1) that generates a single sample $(x, y) \sim \mathcal{D}$; technically, this algorithm runs in at most $10(n+1)$ *expected* number of steps, but as we will see this is sufficient to complete the proof. For ease of notation, we let $\mathcal{Z} := \mathcal{X} \times \mathcal{Y}$ and we identify $\mathcal{Z}$ with $\{0,1\}^{n+1}$ and denote $z \in \mathcal{Z}$ as $(z_1, \ldots, z_{n+1}) = (y, x_1, \ldots, x_n)$. SAMPLE-EXTRACT operates by sampling the bits of $z$ one by one, drawing $\widehat{z}_i$ from the conditional distribution $\{z_i \mid z_{1,\ldots,i-1}\}_{\mathcal{D}}$. We show two things: (i) Once the algorithm is at a prefix $s$, the algorithm indeed returns a sample $\widehat{z}$ drawn from the conditional distribution $\{z \mid z_{1,\ldots,k} = s\}_{\mathcal{D}}$, and (ii) The algorithm returns a sample in $T \leq 10(n+1)$ expected number of steps.

We show (i) by a reverse induction on the length of $s$. The base case of $|s| = n+1$ is trivial. For any $\ell$, observe that conditioned on $v_\ell = 1/b$, the sample returned is indeed sampled according to $\{z \mid z_{1,\ldots,\ell} = s\}_{\mathcal{D}}$. Now, consider conditioning on $v_\ell = t/b$ for some $t \geq 2$. It is easy to see that $v_{\ell+1}/v_\ell$ is distributed according to the (normalized) Binomial distribution $\mathrm{Bin}(t, p)/t$ where $p = \Pr_{\mathcal{D}}[z_{\ell+1} = 1 \mid z_{1,\ldots,\ell} = s]$ and hence $s$ is appended with 1 with probability $\mathbb{E}[v_{\ell+1}/v_\ell \mid v_\ell = t/b] = p$, or appended with 0 otherwise. The inductive hypothesis for $|s| = \ell + 1$ completes the argument.

We show (ii) by proving a couple of more general claims. First of all, we assert that if the algorithm currently has a prefix $s$ that occurs in a random sample from $\mathcal{D}$ with probability $p_s$ and $bp_s \leq 1/5$ then the expected number remaining steps before the algorithm terminates is at most $5/bp_s$. We prove this by reverse induction on the length of $s$. The base case of $|s| = n+1$ is trivial. Now, let $b_s$ be the number of samples in the next batch that start with $s$. We have $\mathbb{E}[b_s] = bp_s$ and

$$\mathbb{E}[\max(0, b_s - 1)] \leq \mathbb{E}[b_s(b_s - 1)] = b(b-1)p_s^2 \leq bp_s/5$$

And hence,

$$\mathbb{P}[b_s = 1] = \mathbb{E}[b_s] - \mathbb{E}[\max(0, b_s - 1)] \geq (4/5)bp_s$$

Thus, the expected number of steps before we get at least one sample starting with $s$ is at most $2/bp_s$. Also, $\Pr[b_s > 0] = 1 - (1 - p_s)^b \leq bp_s$. So, for $p_0 := p_{s \circ 0}$ and $p_1 := p_{s \circ 1}$ (that is, the probabilities that a sample from $\mathcal{D}$ starts with $s \circ 0$ and $s \circ 1$ respectively), the expected number of steps remaining in the algorithm is at most

$$\frac{2}{bp_s} + \frac{\mathbb{P}[b_s > 1]}{\mathbb{P}[b_s > 0]} \left( \frac{p_0}{p_s} \cdot \frac{5}{bp_0} + \frac{p_1}{p_s} \cdot \frac{5}{bp_1} \right) = \frac{2}{bp_s} + \frac{\mathbb{P}[b_s > 1]}{\mathbb{P}[b_s > 0]} \cdot \frac{10}{bp_s} \leq \frac{2}{bp_s} + \frac{1}{4} \cdot \frac{10}{bp_s} \leq \frac{5}{bp_s}$$

as claimed. Next, we show that if the algorithm currently has a prefix $s$ and $bp_s > 1/5$, then the expected number of steps remaining is at most $10(n + 1 - |s|)$. We again prove this with a reverse induction on $|s|$. The base case of $|s| = n+1$ is trivial. With $bp_s > 1/5$, the probability that a batch of $b$ samples has at least one sample starting with prefix $s$ is at least $1/8$. So, the expected remaining number of steps before the algorithm terminates is at most

$$8 + \left( \frac{p_0}{p_s} \cdot \left( 10(n - |s|) + \frac{5}{bp_0} \right) + \frac{p_1}{p_s} \cdot \left( 10(n - |s|) + \frac{5}{bp_1} \right) \right)$$

$$= 8 + (10(n - |s|) + \frac{5}{bp_s} + \frac{5}{bp_s}$$

$$\leq 10(n + 1 - |s|)$$

as desired. That completes the induction argument. The algorithm starts with $s = \epsilon$, which every sample will start with, so the expected number of steps before the algorithm terminates is at most $10(n + 1)$.

Finally, given a PAC$(m, r)$ method $\mathcal{A}$, we design a bSQ$(k, \tau', b', p = n + 1, r')$ method $\mathcal{A}'$ that runs SAMPLE-EXTRACT for $k = 10m(n + 1)/\delta$ rounds, restarting the algorithm after every sample returned. If the total number of samples extracted is less than $m$, then $\mathcal{A}'$ outputs the zero predictor. Else it returns the output of $\mathcal{A}$ on the first $m$ samples extracted. Since the expected number of rounds needed to extract $m$ samples is at most $10m(n + 1)$, we have by Markov's inequality that the probability of extracting less than $m$ samples in $10m(n + 1)/\delta$ rounds is at most $\delta$. Thus, we get that for any $\mathcal{D}$, $\mathrm{err}(\mathcal{A}', \mathcal{D}) \le \mathrm{err}(\mathcal{A}; \mathcal{D}) + \delta$ (where $\mathcal{A}'$ return the null (zero) predictor if less than $m$ samples were extracted, in which case, the loss is 1). The number of random bits needed per round is at most $\log_2 b$ and thus, the total number of random bits needed is $r' = r + k \log_2 b$. $\qquad\square$

**Theorem 2b.** (bSQ to PAC) *For all $b$, $\boldsymbol{\tau}$ and $k, p, r$, it holds that*

$$\mathrm{PAC}(m' = kb, r' = r) \preceq_0 \mathrm{bSQ}(k, \boldsymbol{\tau}, \boldsymbol{b}, p, r).$$

*Furthermore, for every runtime* TIME *it holds for* TIME$' = \mathrm{poly}(n, b, \mathrm{TIME})$ *that*

$$\mathrm{PAC}_{\mathrm{TM}}(m' = kb, r' = r, \mathrm{TIME}') \preceq_0 \mathrm{bSQ}_{\mathrm{TM}}(k, \boldsymbol{\tau}, \boldsymbol{b}, p, r, \mathrm{TIME}).$$

*Proof.* This is immediate, since a PAC$(m = kb, r)$ method can generate valid bSQ responses using $b$ samples for each of the $k$ rounds by simply computing the empirical averages per batch. The number of random bits used remains unchanged. $\qquad\square$

Finally we show that with a slight modification, SAMPLE-EXTRACT can be implemented as a bSQ$^{0/1}$ algorithm, thereby proving Lemma 1, restated below for convenience.

**Lemma 1.** (PAC to bSQ$^{0/1}$) *For all $\delta > 0$, for all $b$, and $\boldsymbol{\tau} < 1/(2b)$, then and for all $m, r$, it holds for $k' = 20m(n + 1)/\delta$, $p' = n + 1$, $r' = r + k \log_2 b$ that*

$$\mathrm{bSQ}^{0/1}(k', \boldsymbol{\tau}, \boldsymbol{b}, p', r') \preceq_\delta \mathrm{PAC}(m, r).$$

*Furthermore, for every runtime* TIME *it holds for* TIME$' = \mathrm{poly}(n, m, r, \mathrm{TIME}, 1/\delta)$ *that*

$$\mathrm{bSQ}^{0/1}_{\mathrm{TM}}(k', \boldsymbol{\tau}, \boldsymbol{b}, p', r', \mathrm{TIME}') \preceq_\delta \mathrm{PAC}_{\mathrm{TM}}(m, r, \mathrm{TIME}).$$

*Proof.* Since $z_1 = y$, all queries of SAMPLE-EXTRACT are already of the form $\mathbb{1}\{y = \overline{y}\} \wedge \Phi_{\mathcal{X}}(x)$ for $\overline{y} \in \{0, 1\}$. It can also be implemented as a bSQ$^{0/1}$ algorithm as follows: After the first query we fix $y = s_1 \in \{0, 1\}$. If $s_1 = 0$, we use only even rounds to perform the queries as done by SAMPLE-EXTRACT and when $s_1 = 1$, we use only odd rounds. This increases the total number of rounds by a factor of 2. $\qquad\square$

## A.2   fbSQ versus PAC

We show the analogs of Theorems 2a and 2b for fbSQ.

**Lemma 5a.** (PAC to fbSQ) *For all $m$, and $\boldsymbol{\tau} < 1/(2m)$ and for all $r$, it holds that*

$$\mathrm{fbSQ}(k = m(n + 1), \tau, \boldsymbol{m'} = m, p = 1, r' = r) \preceq_0 \mathrm{PAC}(m, r).$$

*Furthermore, for every runtime* TIME*, it holds for* TIME$' = \mathrm{poly}(n, m, r, \mathrm{TIME}, 1/\delta)$ *and same $k, p, r'$ as above that*

$$\mathrm{fbSQ}_{\mathrm{TM}}(k, \tau, m' = m, p, r', \mathrm{TIME}') \preceq_0 \mathrm{PAC}_{\mathrm{TM}}(m, r, \mathrm{TIME}).$$

*Proof.* This proof is similar to, but significantly simpler than, the proof of Theorem 2a. Using same notations as in proof of Theorem 2a, for any prefix $s \in \{0, 1\}^\ell$, let $S_s = \{z \in S \mid z_{1,\dots,\ell} = s\}$. We show using reverse induction on $\ell$, that for any given prefix $s$ of length $\ell$ and knowledge of $|S_s| > 0$, we can deterministically recover all samples matching the prefix $s$ using $|S_s| \cdot (n + 1 - \ell)$ many fbSQs.

The base case of $\ell = n + 1$ is trivial, since we already know $|S_s|$. For any $s$, we issue the fbSQ $\Phi : \mathcal{Z} \to \{0, 1\}$ given as $\Phi(z) = \mathbb{1}\{z_{1,\dots,\ell} = s \text{ and } z_{\ell+1} = 1\}$. Since $\tau < 1/2m$, rounding any

value $v$ such that $|v - \mathbb{E}_S \Phi(z)| \leq \tau$ to the nearest integral multiple of $1/m$ gives us $|S_{s\circ 1}|/m$. Thus, with this one query we recover both $|S_{s\circ 1}|$ and $|S_{s\circ 0}| = |S_s| - |S_{s\circ 1}|$. By the inductive assumption, we can recover all samples in $S_{s\circ 0}$ using $|S_{s\circ 0}|(n-\ell)$ additional queries and similarly, all samples in $S_{s\circ 1}$ using $|S_{s\circ 1}|(n-\ell)$ queries. Thus, we recover all samples in $S_s$ using $1 + |S_s|(n-\ell) \leq |S_s|(n+1-\ell)$ queries (since $1 \leq |S_s|$).

Starting with the prefix $s = \epsilon$ (empty string) and knowledge of $|S_\epsilon| = m$, we can recover all samples using at most $m(n+1)$ fbSQs, after which we can simply simulate the PAC$(m, r)$ method. Note that unlike the reduction to bSQ, here the algorithm always succeeds in extracting $m$ samples in $m(n+1)$ steps. Hence there is no loss in the error ensured. $\square$

**Lemma 5b.** (fbSQ to PAC) *For all $m$, $\tau$ and $k, p, r$, it holds that*
$$\mathsf{PAC}(m, r) \preceq_0 \mathsf{fbSQ}(k, \tau, m, p, r).$$
*Furthermore, for every runtime* TIME *it holds for* $\text{TIME}' = \text{poly}(n, m, \text{TIME})$ *that*
$$\mathsf{PAC_{TM}}(m, r, \text{TIME}') \preceq_0 \mathsf{fbSQ_{TM}}(k, \tau, m, p, r, \text{TIME}).$$

*Proof.* This is immediate, since a PAC$(m, r)$ method can generate valid fbSQ responses using $m$ samples for each of the $k$ rounds by simply computing the empirical averages over the entire batch of samples. The number of random bits used remains unchanged. $\square$

Finally, we show that with a slight modification, in the same regime of Lemma 5a, any PAC method can be simulated by a fbSQ$^{0/1}$ method, analogous to Lemma 1.

**Lemma 6.** (PAC to fbSQ$^{0/1}$) *For all $m$, and $\tau < 1/(2m)$, and for all $r$, it holds that*
$$\mathsf{fbSQ}^{0/1}(k = 2m(n+1), \tau, m' = m, p = 1, r' = r) \preceq_0 \mathsf{PAC}(m, r).$$
*Furthermore, for every runtime* TIME, *it holds for* $\text{TIME}' = \text{poly}(n, m, r, \text{TIME}, 1/\delta)$ *and same $k, p, r'$ as above that*
$$\mathsf{fbSQ}^{0/1}_{\mathsf{TM}}(k, \tau, m', p, r', \text{TIME}') \preceq_0 \mathsf{PAC_{TM}}(m, r, \text{TIME}).$$

*Proof.* By modifying proof of Lemma 5a, analogous to the modification to proof of Theorem 2a to get Lemma 1. $\square$

## A.3 bSQ versus SQ

**Theorem 2c.** (bSQ to SQ) *There exists a constant $C \geq 0$ such that for all $\delta > 0$, for all $k$, $\tau$, $b$, $p$, $r$, such that $b\tau^2 > C \log(kp/\delta)$, it holds that*
$$\mathsf{SQ}(k' = kp, \tau' = \frac{\tau}{2}, r' = r) \preceq_\delta \mathsf{bSQ}(k, \tau, b, p, r).$$
*Furthermore, for any runtime* TIME *it holds for* $\text{TIME}' = \text{poly}(\text{TIME})$ *that*
$$\mathsf{SQ_{TM}}(k' = kp, \tau' = \frac{\tau}{2}, r' = r, \text{TIME}') \preceq_\delta \mathsf{bSQ_{TM}}(k, \tau, b, p, r, \text{TIME}).$$

*Proof.* Fix a bSQ$(k, \tau, b, p, r)$ method $\mathcal{A}$, and consider any bSQ query $\Phi : \mathcal{X} \times \mathcal{Y} \to [-1, 1]^p$. Using Chernoff-Hoeffding's bound and a union bound over $p$ entries, we have that
$$\Pr_{S \sim \mathcal{D}^b} \left[ \| \mathbb{E}_S \Phi(x, y) - \mathbb{E}_{\mathcal{D}} \Phi(x, y) \|_\infty > \eta \right] \leq 2pe^{-\eta^2 b/2} \tag{10}$$
Conditioned on $\| \mathbb{E}_S \Phi(x, y) - \mathbb{E}_{\mathcal{D}} \Phi(x, y) \|_\infty \leq \eta$, any response $v \in [-1, 1]^p$ that satisfies $\| v - \mathbb{E}_{\mathcal{D}} \Phi(x, y) \|_\infty \leq \tau - \eta$, also satisfies that $\| v - \mathbb{E}_S \Phi(x, y) \|_\infty \leq \tau$.

Consider a SQ$(k' = kp, \tau' = \tau - \eta, r)$ method $\mathcal{A}'$, that makes the same set of queries as $\mathcal{A}$ (making $p$ SQ queries sequentially for each bSQ query), pretending that the SQ responses received are in fact valid bSQ responses. By a union bound over the $m$ rounds, with probability at least $1 - 2kpe^{-\eta^2 b/2}$, all valid SQ responses to $\mathcal{A}'$ are also valid bSQ responses to $\mathcal{A}$ and hence the statistical distance between the output distribution of $\mathcal{A}'$ and $\mathcal{A}$ is at most $2kpe^{-\eta^2 b/2}$. Hence $\text{err}(\mathcal{A}', \mathcal{D}) \leq \text{err}(\mathcal{A}; \mathcal{D}) + 4kpe^{-\eta^2 b/2}$; here we assume w.l.o.g. that the range of the predictor returned by the bSQ method is $[-1, 1]$, in which case the maximum squared loss of a predictor is at most 2. When $b\tau^2 \geq 8 \log(4kp/\delta)$, setting $\eta = \tau/2$ completes the proof. $\square$

**Theorem 2d.** (SQ to bSQ) *There exists a constant $C \geq 0$ such that for all $\delta > 0$, **for all $b$** and all $k, \tau, r$, it holds that*

$$\mathsf{bSQ}\left(k' = k \cdot \left\lceil \frac{C\log(k/\delta)}{b\tau^2} \right\rceil, \tau' = \frac{\tau}{2}, b, p' = 1, r' = r\right) \preceq_\delta \mathsf{SQ}(k, \tau, r).$$

*Furthermore, for every runtime TIME it holds for $\mathrm{TIME}' = \mathrm{poly}(\mathrm{TIME}, k, 1/\tau)$ and the same $k', \tau', p', r'$, that*

$$\mathsf{bSQ}_{\mathsf{TM}}\left(k', \tau', b, p', r', \mathrm{TIME}'\right) \preceq_\delta \mathsf{SQ}_{\mathsf{TM}}(k, \tau, r, \mathrm{TIME}).$$

*Proof.* For any $\mathsf{SQ}(k, \tau, r)$ method $\mathcal{A}$, consider a $\mathsf{bSQ}(k' = kq, \tau' = \tau/2, b, p = 1, r)$ method $\mathcal{A}'$ that makes the same set of queries as $\mathcal{A}$, but repeating each query $q$ times, and then averaging the $q$ bSQ responses received and treating the average as a valid SQ response ($q$ to be specified shortly). Suppose $v_1, \ldots, v_q$ are bSQ responses, that is, $|v_i - \mathbb{E}_{S_i} \Phi(x, y)| \leq \tau'$ for each $i$, then $v = \sum_{i=1}^q v_i/q$ and $S = \bigcup_i S_i$ satisfies $|v - \mathbb{E}_S \Phi(x, y)| \leq \tau'$, by triangle inequality.

From Equation (10) and a union bound over the $k$ queries, we have that with probability at least $1 - 2ke^{-\eta^2 bq/2}$, any valid bSQ responses $v_1, \ldots, v_q$ for the $q$ queries corresponding to each SQ query, satisfies $|v - \mathbb{E}_\mathcal{D} \Phi(x, y)| \leq \tau' + \eta$. Setting $\eta = \tau - \tau' = \tau/2$, we get that the statistical distance between the output distribution of $\mathcal{A}'$ and $\mathcal{A}$ is at most $2ke^{-\tau^2 bq/8}$. Hence $\mathrm{err}(\mathcal{A}', \mathcal{D}) \leq \mathrm{err}(\mathcal{A}; \mathcal{D}) + 4ke^{-\tau^2 bq/8}$; again, we assume w.l.o.g. that the range of the predictor returned by the SQ method is $[-1, 1]$, in which case the maximum squared loss of a predictor is at most 2. Choosing $q = \left\lceil \frac{8\log(4k/\delta)}{b\tau^2} \right\rceil$ completes the proof. $\qquad\square$

Finally we show that any SQ method can be simulated by a $\mathsf{bSQ}^{0/1}$ method thereby proving Lemma 2, restated below for convenience. The proof goes via an intermediate $\mathsf{SQ}^{0/1}$ method (defined analogous to $\mathsf{bSQ}^{0/1}$).

**Lemma 2.** (SQ to $\mathsf{bSQ}^{0/1}$) *There exists a constant $C > 0$, such that for all $\delta > 0$, **for all $b$** and all $k, \tau, r$, it holds that*

$$\mathsf{bSQ}^{0/1}\left(k' = k \cdot \left\lceil \frac{C\log(k/\delta)}{b\tau^2} \right\rceil, \tau' = \frac{\tau}{4}, b, p' = 1, r' = r\right) \preceq_\delta \mathsf{SQ}(k, \tau, r).$$

*Furthermore, for every runtime TIME. it holds for $\mathrm{TIME}' = \mathrm{poly}(\mathrm{TIME}, k, 1/\tau)$ and the same $k', \tau', p', r'$, that*

$$\mathsf{bSQ}_{\mathsf{TM}}^{0/1}(k', \tau', b, p', r', \mathrm{TIME}') \preceq_\delta \mathsf{SQ}_{\mathsf{TM}}(k, \tau, r, \mathrm{TIME}).$$

*Proof.* Consider any $\mathsf{SQ}(k, \tau, r)$ method $\mathcal{A}$. Let $\Phi_t : \mathcal{X} \times \mathcal{Y} \to [-1, 1]$ be the query issued by the method $\mathcal{A}$. Let $\mathcal{A}'$ be the following $\mathsf{SQ}^{0/1}(2k, \tau/2, r)$ method: In rounds $2t - 1$ and $2t$, $\mathcal{A}'$ issues queries $\Phi_t^1, \Phi_t^0$ respectively, where $\Phi_t^1(x, y) := y \cdot \Phi(x, y)$ and $\Phi_t^0(x, y) := (1 - y) \cdot \Phi(x, y)$. For any valid responses $v_1, v_0$ to the queries $\Phi_t^1, \Phi_t^0$, that is, $|v_1 - \mathbb{E}_\mathcal{D} \Phi_t^1(x, y)| \leq \tau/2$ and $|v_0 - \mathbb{E}_\mathcal{D} \Phi_t^0(x, y)| \leq \tau/2$, it holds by triangle inequality that $|v_0 + v_1 - \mathbb{E}_\mathcal{D}(\Phi_t^0(x, y) + \Phi_t^1(x, y))| \leq \tau$. Thus, $v := v_0 + v_1$ is a valid response for the SQ query $\Phi_t = \Phi_t^1 + \Phi_t^0$. Thus, we get

$$\mathsf{SQ}^{0/1}(2k, \tau/2, r) \preceq_0 \mathsf{SQ}(k, \tau, r).$$

Finally, we use essentially the same argument as in Theorem 2d to obtain a $\mathsf{bSQ}^{0/1}$ method from $\mathcal{A}'$. The only change needed to preserve the alternating nature of $\mathcal{A}'$ is that we perform all the 1-queries in $q$ odd rounds, interleaved with 0-queries in $q$ even rounds. This completes the proof. $\qquad\square$

## A.4 fbSQ versus SQ

We show the analogs of Theorems 2c and 2d for fbSQ. The number of samples required depends linearly on the number of queries, instead of logarithmically. The reason for this is that unlike in the proof of Theorems 2c and 2d, a naive union bound does not suffice, since the queries can be adaptive.

**Lemma 5c.** (fbSQ to SQ) *There exist a constant $C \geq 0$ such that for all $\delta > 0$, **for all** $k, \tau, m, p, r$ such that $m\tau^2 > C(kp\log(1/\tau) + \log(1/\delta))$, it holds that*

$$\mathsf{SQ}(k' = kp, \tau' = \frac{\tau}{2}, r' = r) \preceq_\delta \mathsf{fbSQ}(k, \tau, m, p, r).$$

*Furthermore, for any runtime $\textsc{Time}$ it holds for $\textsc{Time}' = \mathrm{poly}(\textsc{Time})$ that*

$$\mathsf{SQ}_{\mathsf{TM}}(k' = kp, \tau' = \frac{\tau}{2}, r' = r, \textsc{Time}') \preceq_\delta \mathsf{fbSQ}_{\mathsf{TM}}(k, \tau, m, p, r, \textsc{Time}).$$

*Proof.* Fix a $\mathsf{fbSQ}(k, \tau, m, p, r)$ method $\mathcal{A}$. Conditioned on the choice of randomness sampled by $\mathcal{A}$, consider the set of all possible queries that $\mathcal{A}$ could make, assuming that all the responses to all the queries ever made are in $2\alpha \cdot \mathbb{Z}^p \cap [-1, 1]^p$, that is, each entry of the response is an integral multiple of $2\alpha$ (choice of $\alpha$ to be made later). The number of distinct responses to any query then is at most $(1 + 1/\alpha)^p$. Even accounting for the adaptive nature of the method, conditioned on the choice of randomness, there are at most $(1/\alpha + 1)^{kp}$ possible different transcripts of queries and answers, and hence a total of at most $(1/\alpha + 1)^{kp}$ distinct queries that the method could have made.

Using Chernoff-Hoeffding's bound (Equation (10)) and a union bound over all these $(1/\alpha + 1)^{kp}$ possible queries we have that with probability at least $1 - 2p(1 + 1/\alpha)^{kp} \cdot e^{-\eta^2 m/2}$ over sampling $S \sim \mathcal{D}^m$, it holds for each such query $\Phi : \mathcal{X} \times \mathcal{Y} \to [-1, 1]^p$ that

$$\|\mathbb{E}_S\, \Phi(x, y) - \mathbb{E}_\mathcal{D}\, \Phi(x, y)\|_\infty \;<\; \eta$$

Conditioned on $\|\mathbb{E}_S\, \Phi(x, y) - \mathbb{E}_\mathcal{D}\, \Phi(x, y)\|_\infty \leq \eta$, any response $v \in [-1, 1]^p$ that satisfies $\|v - \mathbb{E}_\mathcal{D}\, \Phi(x, y)\|_\infty \leq \tau - \eta - \alpha$, also satisfies that $\|v - \mathbb{E}_S\, \Phi(x, y)\|_\infty \leq \tau - \alpha$. Furthermore, let $\tilde{v}$ be the rounding of $v$ to the nearest value in $2\alpha \cdot \mathbb{Z}^p \cap [-1, 1]^p$. Then, it holds that $\|\tilde{v} - v\|_\infty \leq \alpha$ and hence $\|\tilde{v} - \mathbb{E}_S\, \Phi(x, y)_\infty\| \leq \tau$.

Consider a $\mathsf{SQ}(k' = kp, \tau' = \tau - \eta - \alpha, r)$ method $\mathcal{A}'$, that makes the same set of queries as $\mathcal{A}$ (making $p$ SQ queries sequentially for each fbSQ query), but rounds the SQ responses to the nearest value in $2\alpha \cdot \mathbb{Z}^p \cap [-1, 1]^p$, and treats those as valid fbSQ responses. By the union bound argument presented above, it holds with probability at least $1 - 2p(1 + 1/\alpha)^{kp} \cdot e^{-\eta^2 m/2}$ that the rounded SQ responses are also valid fbSQ responses. Hence, the statistical distance between the output distribution of $\mathcal{A}'$ and $\mathcal{A}$ is at most $2p(1 + 1/\alpha)^{kp} \cdot e^{-\eta^2 m/2}$. Hence $\mathrm{err}(\mathcal{A}', \mathcal{D}) \leq \mathrm{err}(\mathcal{A}, \mathcal{D}) + 4p(1 + 1/\alpha)^{kp} \cdot e^{-\eta^2 m/2}$; here we assume w.l.o.g. that the range of the predictor returned by the fbSQ method is $[-1, 1]$, in which case the maximum squared loss of a predictor is at most 2. When $m\tau^2 \geq 32(kp\log(4/\tau + 1) + \log(4p/\delta))$, setting $\alpha = \eta = \tau/4$ completes the proof. $\square$

**Lemma 5d.** (SQ to fbSQ) *There exist a constant $C \geq 0$ such that for all $\delta > 0$, $k, \tau, r$, it holds for $\tau' = \frac{\tau}{2}$ and all $m$ such that $m\tau^2 > C(k\log(1/\tau) + \log(1/\delta))$ that*

$$\mathsf{fbSQ}(k' = k, \tau', m, p' = 1, r' = r) \preceq_\delta \mathsf{SQ}(k, \tau, r).$$

*Furthermore, for every runtime $\textsc{Time}$ it holds for $\textsc{Time}' = \mathrm{poly}(\textsc{Time}, k, \tau^{-1})$ that*

$$\mathsf{fbSQ}_{\mathsf{TM}}(k' = k, \tau', m, p' = 1, r' = r, \textsc{Time}') \preceq_\delta \mathsf{SQ}_{\mathsf{TM}}(k, \tau, r, \textsc{Time}).$$

*Proof.* Fix an $\mathsf{SQ}(k, \tau, r)$ method $\mathcal{A}$. Similar to the proof of Part 1, conditioned on the choice of randomness sampled by $\mathcal{A}$, consider the set of all possible queries that $\mathcal{A}$ could make, assuming that all the responses to all the queries every made in $2\alpha \cdot \mathbb{Z}^p \cap [-1, 1]$. Similar to before, the number of such queries is at most $(1 + 1/\alpha)^k$.

Consider a $\mathsf{fbSQ}(k' = k, \tau' = \tau/2, m, p = 1, r)$ method $\mathcal{A}'$ that makes the same set of queries as $\mathcal{A}$ (for $\tau'$ to be decided shortly), but rounds the fbSQ responses to the nearest value in $2\alpha \cdot \mathbb{Z} \cap [-1, 1]$ and treats those as valid SQ responses. For any query $\Phi$ if $v$ is a valid fbSQ response, then we have $|v - \mathbb{E}_S\, \Phi(x, y)| \leq \tau'$ and for its rounding $\tilde{v}$, it holds that $|\tilde{v} - \mathbb{E}_S\, \Phi(x, y)| \leq \tau' + \alpha$. From a similar application of Chernoff-Hoeffding's bound and a union bound, we have that with probability at least $1 - 2(1 + 1/\alpha)^k \cdot e^{-\eta^2 m/2}$ that for all the queries $\Phi : \mathcal{X} \times \mathcal{Y} \to [-1, 1]$ considered above, and for any valid fbSQ response $v$, it holds that $|v - \mathbb{E}_\mathcal{D}\, \Phi(x, y)| \leq \tau' + \alpha + \eta$. Hence, we get that the statistical distance between the output distribution of $\mathcal{A}'$ and $\mathcal{A}$ is at most $2(1 + 1/\alpha)^k \cdot e^{-\eta^2 m/2}$

and hence $\mathrm{err}(\mathcal{A}', \mathcal{D}) \leq \mathrm{err}(\mathcal{A}; \mathcal{D}) + 4(1 + 1/\alpha)^k \cdot e^{-\eta^2 m/2}$; again, we assume w.l.o.g. that the range of the predictor returned by the SQ method is $[-1, 1]$, in which case the maximum squared loss of a predictor is at most 2. Finally, setting $\alpha = \frac{\tau}{4}$ and $\eta = \frac{\tau}{4}$, the result holds for $m \geq \left\lceil \frac{32(k \log(4/\tau+1)+\log(4/\delta))}{\tau^2} \right\rceil$. $\qquad\square$

Finally, we show that with a slight modification, in the same regime of Lemma 5d, any SQ method can be simulated by a fbSQ$^{0/1}$ method, analogous to Lemma 2.

**Lemma 7.** (SQ to fbSQ$^{0/1}$) *There exist a constant $C \geq 0$ such that for all $\delta > 0$, $k, \tau, r$, it holds for* $\boldsymbol{\tau' = \frac{\tau}{4}}$ *and all $\boldsymbol{m}$ such that $\boldsymbol{m\tau^2 > C(k \log(1/\tau) + \log(1/\delta))}$ that*

$$\mathsf{fbSQ}(k' = 2k, \tau', m, p' = 1, r' = r) \preceq_\delta \mathsf{SQ}(k, \tau, r).$$

*Furthermore, for every runtime* TIME *it holds for* TIME$' = \mathrm{poly}(\text{TIME}, k, \tau^{-1})$ *that*

$$\mathsf{fbSQ}_{\mathsf{TM}}(k' = 2k, \tau', m, p' = 1, r' = r, \text{TIME}') \preceq_\delta \mathsf{SQ}_{\mathsf{TM}}(k, \tau, r, \text{TIME}).$$

*Proof.* By modifying proof of Lemma 5d, analogous to the modification to proof of Theorem 2d to get Lemma 2. $\qquad\square$

# B  Simulating bSQ with bSGD : Proof of Lemma 3a

In this section, we show how to simulate any bSQ$^{0/1}$ method $\mathcal{A}$ as bSGD on some differentiable model constructed according to $\mathcal{A}$, and thus prove Lemma 3a:

**Lemma 3a.** (bSQ$^{0/1}$ to bSGD) *For all $\tau \in (0,1)$ and $b, k, p, r \in \mathbb{N}$, it holds that*

$$\mathsf{bSGD}\left(T' = k, \rho = \frac{\tau}{4}, b, p' = r + (p+1)k, r' = r\right) \preceq_0 \mathsf{bSQ}^{0/1}(k, \tau, b, p, r).$$

**Simulating a single $\overline{y}$-query.**   As a first step towards showing Lemma 3a, we show how a single $\overline{y}$-query can be simulated using a single step of bSGD. We consider parameterized queries, that can depend on some of parameters of the differentiable model. Namely, $\Phi : [-1, 1]^q \times \mathcal{X} \times \mathcal{Y} \to [-1, 1]^p$ is a query that given some parameters $\theta \in [-1, 1]^q$, an input $x \in \mathcal{X}$ and a label $y \in \mathcal{Y}$, returns some vector value $\Phi(\theta, x, y)$. We show the following:

**Lemma 8.** *Let $\Phi : [-1,1]^q \times \mathcal{X} \times \mathcal{Y} \to [-1,1]^p$ be some $\overline{y}$-query that is differentiable w.r.t. its parameters. Fix some batch size $b \in \mathbb{N}$, precision $\rho > 0$ and some $\varepsilon > 0$. Then, there exists a differentiable model $f_{\boldsymbol{w}}$, with $\boldsymbol{w} = (\widehat{\theta}; \theta; \kappa) \in \mathbb{R}^q \times \mathbb{R}^p \times \mathbb{R}$, such that running bSGD for $T = 1$ step, from an initialization $\boldsymbol{w}^{(0)} = (\widehat{\theta^{(0)}}, \theta^{(0)}, \kappa^{(0)})$ satisfying $\theta^{(0)}, \kappa^{(0)} = 0$, yields parameters $\boldsymbol{w}^{(1)} = (\widehat{\theta}^{(1)}; \theta^{(1)}; \kappa^{(1)}) \in \mathbb{R}^q \times \mathbb{R}^p \times \mathbb{R}$ such that for $S \sim \mathcal{D}^b$:*

1. $\left\| \theta^{(1)} - \frac{1}{b} \sum_{(x,y) \in S} \Phi(\widehat{\theta^{(0)}}, x, y) \right\|_\infty \leq \varepsilon + \rho,$

2. $\kappa^{(1)} \geq \varepsilon - \rho.$

*Proof.* We consider two cases:

1. Assume $\Phi$ is a 0-query, namely $\Phi(x, y) = (1 - y)\Phi(\widehat{\theta}, x, 0)$. Then, we define a differentiable model as follows:
$$f_{\boldsymbol{w}}(x) = 1 - \left\langle \Phi(\widehat{\theta}, x, 0), \theta \right\rangle - \kappa + \varepsilon$$

   Fix some batch $S$, and observe that:
$$\frac{\partial}{\partial \theta} \mathbb{E}_S \ell_{\mathrm{sq}}(f_{\boldsymbol{w}}(x), y) = \mathbb{E}_S(f_{\boldsymbol{w}}(x) - y)\frac{\partial}{\partial \theta} f_{\boldsymbol{w}}(x) = -\mathbb{E}_S(1 + \varepsilon - y)\Phi(\widehat{\theta}, x, 0)$$
$$= -\mathbb{E}_S \Phi(\widehat{\theta}, x, y) - \varepsilon \mathbb{E}_S \Phi(\widehat{\theta}, x, 0)$$

   Then, after performing one step of bSGD we have:
$$\left\| \theta^{(1)} - \mathbb{E}_S \Phi(\widehat{\theta}, x, y) \right\|_\infty \leq \varepsilon + \rho$$

Now, similarly we have:

$$\frac{\partial}{\partial \kappa} \mathbb{E}_S \ell_{\text{sq}}(f_{\boldsymbol{w}}(x), y) = \mathbb{E}_S(f_{\boldsymbol{w}}(x) - y)\frac{\partial}{\partial \kappa} f_{\boldsymbol{w}}(x) = -\mathbb{E}_S(1 + \varepsilon - y) \leq -\varepsilon$$

And therefore, after one step of bSGD we have $\kappa^{(1)} \geq \varepsilon - \rho$.

2. Assume $\Phi$ is a 1-query, namely $\Phi(x, y) = y\Phi(\widehat{\theta}, x, 0)$. Then, we define a differentiable model as follows:

$$f_{\boldsymbol{w}}(x) = \left\langle \Phi(\widehat{\theta}, x, 1), \theta \right\rangle + \kappa - \varepsilon$$

Fix some batch $S$, and observe that:

$$\frac{\partial}{\partial \theta} \mathbb{E}_S \ell_{\text{sq}}(f_{\boldsymbol{w}}(x), y) = \mathbb{E}_S(f_{\boldsymbol{w}}(x) - y)\frac{\partial}{\partial \theta} f_{\boldsymbol{w}}(x) = \mathbb{E}_S(-\varepsilon - y)\Phi(\widehat{\theta}, x, 1)$$

$$= -\mathbb{E}_S \Phi(\widehat{\theta}, x, y) - \varepsilon\mathbb{E}_S \Phi(\widehat{\theta}, x, 0)$$

Then, after performing one step of bSGD we have:

$$\left\| \theta^{(1)} - \mathbb{E}_S \Phi(\widehat{\theta}, x, y) \right\|_\infty \leq \varepsilon + \rho$$

Now, similarly we have:

$$\frac{\partial}{\partial \kappa} \mathbb{E}_S \ell_{\text{sq}}(f_{\boldsymbol{w}}(x), y) = \mathbb{E}_S(f_{\boldsymbol{w}}(x) - y)\frac{\partial}{\partial \kappa} f_{\boldsymbol{w}}(x) = \mathbb{E}_S(-\varepsilon - y) \leq -\varepsilon$$

And therefore, after one step of bSGD we have $\kappa^{(1)} \geq \varepsilon - \rho$. $\qquad \square$

**Simulating a bSQ$^{0/1}$ method.** Our goal is to use Lemma 8 to simulate a bSQ$^{0/1}$ method. Any bSQ$^{0/1}$ method $\mathcal{A}$ is completely described by a sequence of (potentially adaptive) queries $\Phi_1, \ldots, \Phi_T$, and a predictor $h$ which depends on the answer to previous queries, namely:

▸ $\Phi_t$ depends on $r$ random bits, and on the answers of the previous $t - 1$ queries, namely:

$$\Phi_t : \{0, 1\}^r \times [-1, 1]^{p \times (t-1)} \times \mathcal{X} \times \mathcal{Y} \to [-1, 1]^p$$

▸ $\Phi_t$ is a $(t \bmod 2)$-query.

▸ $h : \{0, 1\}^r \times [-1, 1]^{p \times (t-1)} \times \mathcal{X} \to \mathbb{R}$ is a predictor that depends on a sequence of random bits denoted $v_0 \in \{0, 1\}^r$, and on the answers to all previous queries, denoted $v_1, \ldots, v_T \in [-1, 1]^p$.

For $v \in \mathbb{R}^q$, let $\langle v \rangle_\tau$ denote the entry-wise rounding of $v$ to $\tau\mathbb{Z}$, namely $\langle v \rangle_\tau := \arg\min_{v' \in (\tau\mathbb{Z})^q} \|v - v'\|_\infty$. In order to prove Lemma 3a, we need the following technical lemma:

**Lemma 9.** *Let* $\Phi : [-1, 1]^q \to [-1, 1]^p$ *be some function, and let* $\delta \in \mathbb{R}$ *be some accuracy. Then, there exists a smooth function* $\tilde{\Phi} : [-1, 1]^q \to [-1, 1]^p$ *such that* $\tilde{\Phi}(x) = \Phi(\langle x \rangle_\delta)$ *for every* $x$ *such that* $\|x - \langle x \rangle_\delta\|_\infty \leq \frac{\delta}{4}$.

We use the following fact:

**Fact 1.** *For every compact set* $K$ *and open set* $U$ *such that* $K \subseteq U \subseteq [-1, 1]^q$, *there exists a smooth function* $\Psi : [-1, 1]^q \to [-1, 1]$ *such that* $\Psi(x) = 1$ *for every* $x \in K$ *and* $\Psi(x) = 0$ *for every* $x \notin U$.

*Proof of Lemma 9.* Now, for some $x \in [-1, 1]^q$ we define $K_x := \times_{i=1}^q [x_i - \delta/4, x_i + \delta/4]$, and $U_x := \times_{i=1}^q (x_i - \delta/3, x_i + \delta/3)$ and note that $K_x$ is compact, $U_x$ is open and $K_x \subseteq U_x$. So, using the above fact, there exists a smooth $\Psi_x$ such that $\Psi_x(K_x) = 1$ and $\Psi_x(\mathbb{R}^q \setminus U_x) = 0$. Now, consider $\tilde{\Phi} : \mathbb{R}^q \to \mathbb{R}^p$ such that

$$\tilde{\Phi}(x) := \sum_{x' \in (\delta\mathbb{Z})^q} \Psi_{x'}(x) \cdot \Phi(x').$$

Whenever $\|x - \langle x \rangle_\delta\|_\infty \leq \delta/4$, it holds that $\Psi_{\langle x \rangle_\delta}(x) = 1$ and $\Psi_{x'}(x) = 0$ for all $x' \in (\delta\mathbb{Z})^q \setminus \{\langle x \rangle_\delta\}$, and hence $\tilde{\Phi}(x) = \Phi(\langle x \rangle_\delta)s$, whenever $\|x - \langle x \rangle_\delta\|_\infty \leq \delta/4$. Finally, since $\Psi_x$ is smooth, $\tilde{\Phi}$ is also differentiable. $\qquad \square$

**Proof of Lemma 3a.** Let $\mathcal{A}$ be a $\mathsf{bSQ}^{0/1}(m, \tau, b, p, r)$ method. Let $\mathcal{A}_R(S_1, \ldots, S_T)$ denote the set of predictors returned by $\mathcal{A}$ after receiving valid responses for $\tau$-precision bSQs on mini-batches $S_1, \ldots, S_T$, using the random bits $R$. In order to prove Lemma 3a, it suffices to show that there exists a differentiable model $f_{\boldsymbol{w}}$, such that, for every choice of $S_1, \ldots, S_T$ of size $b$, and every sequence of bits $R \in \{0, 1\}^r$, there exists an initialization of the model such that bSGD using mini-batches $S_1, \ldots, S_T$ with $\ell_{\mathrm{sq}}$ loss and learning-rate $\gamma = 1$, returns a function $f_{\boldsymbol{w}^{(T)}}$ such that $f_{\boldsymbol{w}^{(T)}} \in \mathcal{A}_R(S_1, \ldots, S_T)$.

Let $\Phi_1, \ldots, \Phi_T$ be the queries made by $\mathcal{A}$, and $h$ be the returned predictor. Using Lemma 9, let $\tilde{\Phi}_1, \ldots, \tilde{\Phi}_T$ be a sequence of queries, with $\tilde{\Phi}_t : [-1, 1]^r \times [-1, 1]^{p \times t-1} \times \mathcal{X} \times \mathcal{Y} \to \mathbb{R}^p$, such that:

- $\tilde{\Phi}_t(v_0, \ldots, v_{t-1}, x, y)$ is smooth w.r.t. $v_0, \ldots, v_{t-1}$.

- $\tilde{\Phi}_t(v_0, \ldots, v_{t-1}, x, y) = \Phi(\langle v_0 \rangle_{\tau/4}, \ldots, \langle v_{t-1} \rangle_{\tau/4}, x, y)$ for $v_0, \ldots v_{t-1}$ satisfying $\left\| v_i - \langle v_i \rangle_{\tau/4} \right\|_\infty \leq \frac{\tau}{32}$ for all $1 \leq i \leq t-1$.

Let $\tilde{h} : [-1, 1]^r \times [-1, 1]^{p \times (t-1)} \times \mathcal{X} \to \mathbb{R}$ be a smooth function that agrees with $h$, defined similarly to $\tilde{\Phi}_t$.

Let $c : \mathbb{R}^3 \to \mathbb{R}$ be a differentiable function such that:

$$c(\alpha_1, \alpha_2, \alpha_3) = \begin{cases} \alpha_3 & \alpha_1 \geq \rho \text{ and } \alpha_2 \leq \rho/2 \\ 0 & \alpha_1, \alpha_2 \geq \rho \\ 0 & \alpha_1, \alpha_2 \leq \rho/2 \\ * & \text{otherwise} \end{cases}$$

Our differentiable model will use the following parameter:

- Parameters $\theta^{(0)} \in \mathbb{R}^p$, initialized to $R$, and stores the "random" bits.

- $T$ sets of parameters, each of size $r$, denoted $\theta^{(1)}, \ldots, \theta^{(T)} \in \mathbb{R}^r$. The parameter $\theta^{(i)}$ will "record" the output of the $i$-th query. We initialize $\theta^{(1)}, \ldots, \theta^{(T)} = 0$.

- $T$ "clock" parameters $\kappa_1, \ldots \kappa_T$, that indicate which query should be issued next. We initialize $\kappa_1, \ldots, \kappa_T = 0$.

We denote by $\theta^{(t,i)}, \kappa_t^{(i)}$ the value of the $t$-th set of parameters in the $i$-th iteration of SGD. The differentiable model is defined as follows:

$$F_{\theta^{(0)}, \ldots, \theta^{(T)}, \kappa_1, \ldots, \kappa_T}(x) = \sum_{t=1}^T c\left(\kappa_{t-1}, \kappa_t, f^{(t)}_{(\theta^{(0)} \ldots \theta^{(t-1)}; \theta^{(t)}; \kappa_t)}(x)\right) + c\left(\kappa_T, 0, \tilde{h}\left(\theta^{(0)}, \ldots, \theta^{(T)}, x\right)\right)$$

Where for every $t$, $f^{(t)}_{(\theta^{(1)} \ldots \theta^{(t-1)}, \theta^{(t)}, \kappa_t)}$ is the differentiable model simulating $\tilde{\Phi}_t$, that is guaranteed by Lemma 8. As a convention, we take $\kappa_0 = 1$ (this is not a trainable parameter of the model).

Now, denote $v_0 := \theta^{(0,0)}$, and for every $t > 0$ denote $v_t := \theta^{(t,t)}$. We have the following claim:

*Claim*: for every iteration $i$ of bSGD,

1. For every $t > i$ we have $\theta^{(t,i)} = 0$ and $\kappa_t^{(i)} = 0$.

2. For $t = i$ we have:

$$\left\| \theta^{(t,i)} - \frac{1}{b} \sum_{(x,y) \in S^{(i)}} \Phi_t\left(v_0, v_1, \ldots, v_{t-1}, x, y\right) \right\|_\infty \leq 3\rho$$

3. For $t < i$ we have $\theta^{(t,i)} = \theta^{(t,t)}$.

4. For $t \leq i$ we have $\kappa_t^{(i)} \geq \rho$.

*Proof*: By induction on $i$:

▸ For $i = 1$, notice that by the initialization, $\kappa_t^{(0)} = 0$ for every $t$. Fix some $t > 1 = i$, and note that $c(\kappa_{t-1}, \kappa_t, \alpha) = 0$, so the gradient w.r.t $\theta^{(t,i)}$, $\kappa_t^{(i)}$ is zero, and so condition 1 hold (the initialization is zero, and the gradient is zero). For $t = 1 = i$, notice that since $c(\kappa_0, \kappa_1^{(0)}, \alpha) = \alpha$, we have:

$$F_{\theta^{(0,0)},\ldots,\theta^{(T,0)},\kappa_1^{(0)},\ldots,\kappa_T^{(0)}}(x) = f_{(\theta^{(0,0)};\theta^{(0,1)};\kappa_1^{(0)})}^{(1)}(x)$$

and by applying Lemma 8 with $\varepsilon = 2\rho$ we get:

$$\left\| \theta^{(1,1)} - \frac{1}{b} \sum_{(x,y) \in S} \tilde{\Phi}_1(\theta^{(0,0)}, x, y) \right\|_\infty \leq \left\| \theta^{(1,0)} \right\|_\infty + \varepsilon + \rho \leq 3\rho$$

and so condition 2 follows from the fact that $v_0 = \theta^{(0,0)} = \langle v_0 \rangle_{\tau/4}$ and so $\tilde{\Phi}_1(v_0, x, y) = \Phi_1(\langle v_0 \rangle_{\tau/4}, x, y) = \Phi_1(v_0, x, y)$. Furthermore, again using Lemma 8 we have:

$$\kappa_1^{(1)} \geq \varepsilon - \rho = \rho$$

and so condition 4 follows. Finally, condition 3 is vacuously true.

▸ Fix some $i > 0$, and assume the claim holds for $i$. We will prove the claim for $i + 1$. By the assumption, we have $\kappa_t^{(i)} = 0$ for every $t > i$ and $\kappa_t^{(i)} \geq \rho$ for every $t \leq i$. Therefore, by definition of $c$, we have $c(\kappa_{t-1}^{(i)}, \kappa_t^{(i)}, \alpha) = \mathbb{1}_{t=i+1}\alpha$. So,

$$F_{\theta^{(0,i)},\ldots,\theta^{(T,i)},\kappa_1^{(i)},\ldots,\kappa_T^{(i)}}(x) = f_{(\theta^{(0,i)},\ldots,\theta^{(i,i)};\theta^{(i+1,i)};\kappa_{i+1}^{(i)})}^{(i+1)}(x)$$

Therefore, conditions 1 and 3 follow from the fact that the gradient with respect to $\theta^{(t,i)}$ and $\kappa_t^{(i)}$, for every $t \neq i + 1$, is zero. Now, using Lemma 8 with $\varepsilon = 2\rho$, condition 2 follows, and we also have $\kappa_{i+1}^{(i+1)} \geq \rho$. Finally, for every $t < i + 1$, by the assumption we have $\kappa_t^{(i)} \geq \rho$, and since the gradient with respect to $\kappa_t^{(i)}$ is zero, we also have $\kappa_t^{(i+1)} \geq \rho$. Therefore, condition 4 follows.

Finally, to prove the Theorem, observe that by the previous claim, for every $t$:

$$\left\| v_t - \frac{1}{b} \sum_{(x,y) \in S^{(2t-1)}} \Phi_t(v_0, \ldots, v_{t-1}, x, y) \right\|_\infty \leq 3\rho < \tau$$

So, $v_t$ is a valid response for the $t$-th query.

By the previous claim, we have $\kappa_t^{(T)} \geq \rho$ for every $1 \leq t \leq T$. Therefore, we have:

$$F_{\theta^{(0,2T)},\ldots,\theta^{(T,T)},\kappa_1^{(T)},\ldots,\kappa_T^{(T)}}(x) = \tilde{h}(\theta^{(0,T)}, \ldots, \theta^{(T,T)})$$

and using the previous claim we have $\theta^{(t',T)} = v_t$ for every $t'$, and therefore, by definition of $\tilde{h}$ we have:

$$F_{\theta^{(0,T)},\ldots,\theta^{(T,T)},\kappa_1^{(T)},\ldots,\kappa_T^{(T)}}(x) = h(v_0, \ldots, v_T)$$

and, using the fact that $v_0, \ldots, v_T$ are valid responses to the method's queries, we get the required. □

## B.1 From Arbitrary Differentiable Models to Neural Networks.

In this section we proved the key lemma for our main results, showing that alternating batch-SQ methods can be simulated by gradient descent over arbitrary differentiable models. We would furthermore like to show that if the alternative batch-SQ method is computationally bounded, the differentiable model we defined can be implemented as a neural network of bounded size.

Indeed, observe that when the method can be implemented using a Turing-machine, each query (denoted by $\Phi$ in the proof) can be simulated by a Boolean circuit [see 5], and hence by a neural network with some fixed weights. Therefore, one can show with little extra effort that the differentiable model introduced in the proof of Lemma 3a can be written as a neural network, with some of the weights being fixed. To show that the same behavior is guaranteed even when all the weights are trained, it is enough to show that all the relevant weights (e.g., $\theta^{(0)}, \ldots, \theta^{(T)}$) have zero gradient, unless they are correctly updated. This can be achieved using the "clock" mechanism (the function $c(\alpha_1, \alpha_2, \alpha_3)$ in the construction), which in turn can be implemented by a neural network that is robust to small perturbations of its weights, and hence does not suffer from unwanted updates of gradient descent. One possible way to implement the clock mechanism using a neural network that is robust to small perturbations is to rely either on large weight magnitudes and small step-sizes, or on the clipping of large gradients.

We do not include these details, Instead, in the next Section, we provide complete details and a rigorous proof of an alternate, more direct, construction of a neural network defining a differentiable model that simulates a given $\mathsf{bSQ}_{\mathsf{TM}}^{0/1}$ method. This direct neural-network construction is based on the same ideas, but is different in implementation from the construction shown in this section, involving some technical details to ensure that the network is well behaved under the gradient descent updates.

## C  Simulating $\mathsf{bSQ_{TM}}$ with $\mathsf{bSGD_{NN}^\sigma}$ : Proof of Lemma 3b

In this section, we show a direct construction of a neural network such that gradient descent on the neural net simulates a given $\mathsf{bSQ}_{\mathsf{TM}}^{0/1}$ method, thus proving Lemma 3b:

**Lemma 3b.** ($\mathsf{bSQ}_{\mathsf{TM}}^{0/1}$ to $\mathsf{bSGD}_{\mathsf{NN}}^\sigma$) *For all $\tau \in (0, \frac{1}{3})$ and $b, k, p, r \in \mathbb{N}$, and using the activation $\sigma$ from Figure 1, it holds that*

$$\mathsf{bSGD}_{\mathsf{NN}}^\sigma \left( T' = k, \rho = \frac{\tau}{4}, b, p' = \mathrm{poly}(k, \frac{1}{\tau}, b, p, r, \mathrm{TIME}), r' = r \right) \preceq_0 \mathsf{bSQ}_{\mathsf{TM}}^{0/1}(k, \tau, b, p, r, \mathrm{TIME}).$$

Given a $\mathsf{bSQ}$ algorithm with a specified bounded runtime, we will design a neural network such that the mini-batch gradients at each step correspond to responses to queries of the $\mathsf{bSQ}$ algorithm. Our proof of this is based on the fact that any efficient $\mathsf{bSQ}_{\mathsf{TM}}^{0/1}$ algorithm must decide what query to make next and what to output based on some efficient computation performed on random bits and the results of previous queries. Any efficient algorithm can be performed by a neural net, and it is possible to encode any circuit as a neural net of comparable size in which every vertex is always at a flat part of the activation function. Doing that would ensure that none of the edge weights ever change, and thus that the net would continue computing the desired function indefinitely. So, that allows us to give our net a subgraph that performs arbitrary efficient computations on the net's inputs and on activations of other vertices.

Also, we can rewrite any $\mathsf{bSQ}_{\mathsf{TM}}^{0/1}$ algorithm to only perform binary queries by taking all of the queries it was going to perform, and querying the $i$th bit of their binary representation for all sufficiently small $i$ instead. For each of the resulting binary queries, we will have a corresponding vertex with an edge going to it from the constant vertex and no other edges going to it. So, the computation subgraph of the net will be able to determine the current weights of the edges leading to the query vertices by checking their activations. Also, each query vertex will have paths from it to the output vertex with intermediate vertices that will either get inputs in the flat parts of their activation function or not depending on the output of some vertices in the computation subgraph. The net effect of this will be to allow the computation subgraph to either make the value encoded by the query edge stay the same or make it increase if the net's output differs from the sample output based on any efficiently computable function of the inputs and other query vertices' activations. This allows us to encode an arbitrary $\mathsf{bSQ}^{\mathrm{fgt}}$ algorithm as a neural net.

**The emulation net.**   In order to prove the capabilities of a neural net trained by batch stochastic gradient descent, we will start by proving that any algorithm in $\mathsf{bSQ}$ can be emulated by a neural net trained by batch stochastic gradient descent under appropriate parameters. In this section our net will use an activation function $\sigma$, as defined in Figure 1, namely

$$\sigma(x) = \begin{cases} -2 & \text{if } x < -3 \\ x+1 & \text{if } -3 \le x \le -1 \\ 0 & \text{if } -1 < x < 0 \\ x & \text{if } 0 \le x \le 2 \\ 2 & \text{if } x > 2 \end{cases}$$

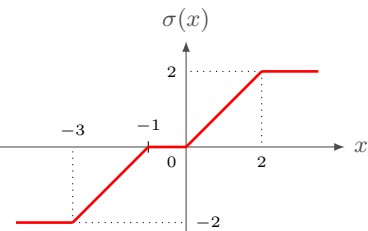

and a loss function of $\ell_{\text{sq}}(y, y') = \frac{1}{2}(y - y')^2$.

Any bSQ algorithm repeatedly makes a query and then computes what query to perform next from the results of the previous queries. So, our neural net will have a component designed so that we can make it update targeted edge weights by an amount proportional to the value of an appropriate query on the current batch and a component designed to allow us to perform computations on these edge weights. We will start by proving that we can make the latter component work correctly. More formally, we assert the following.

**Lemma 10** (Backpropagation-proofed noise-tolerant circuit emulation). *Let $h : \{0,1\}^m \to \{0,1\}^{m'}$ be a function that can be computed by a circuit made of AND, OR, and NOT gates with a total of $b$ gates. Also, consider a neural net with $m$ input[12] vertices $v'_1, ..., v'_m$, and choose real numbers $y_i^{(0)} < y_i^{(1)}$ for each $1 \le i \le m$. It is possible to add a set of at most $b$ new vertices to the net, including output vertices $v''_1, ..., v''_{m'}$, along with edges leading to them such that for any possible addition of edges leading from the new vertices to old vertices, if the net is trained by bSGD, the output of $v'_i$ is either less than $y_i^{(0)}$ or more than $y_i^{(1)}$ for every $i$ in every timestep, then the following hold:*

1. *The derivative of the loss function with respect to the weight of each edge leading to a new vertex is $0$ in every timestep, and no paths through the new vertices contribute to the derivative of the loss function with respect to edges leading to the $v'_i$.*

2. *In any given time step, if the output of $v'_i$ encodes $x_i$ with values less than $y_i^{(0)}$ and values greater than $y_i^{(1)}$ representing $0$ and $1$ respectively for each $i$, then the output of $v''_j$ encodes $h_j(x_1, ..., x_m)$ for each $j$ with $-2$ and $2$ encoding $0$ and $1$ respectively.*

*Proof.* In order to do this, we will add one new vertex for each gate and each input in a circuit that computes $h$. When the new vertices are used to compute $h$, we want each vertex to output $2$ if the corresponding gate or input outputs a $1$ and $-2$ if the corresponding gate or input outputs a $0$, and we want the derivative of its activation with respect to its input to be $0$. In order to do that, we need the vertex to receive an input of more than $2$ if the corresponding gate outputs a $1$ and an input of less than $-3$ if the corresponding gate outputs a $0$.

In order to make one new vertex compute the NOT of another new vertex, it suffices to have an edge of weight $-2$ to the vertex computing the NOT and no other edges to that vertex. We can compute an AND of two new vertices by having a vertex with two edges of weight $2$ from these vertices and an edge of weight $-4$ from the constant vertex. Similarly, we can compute an OR of two new vertices by having a vertex with two edges of weight $2$ from these vertices and an edge of weight $4$ from the constant vertex. For each $i$, in order to make a new vertex corresponding to the $i$th input, we add a vertex and give it an edge of weight $8/(y^{(1)} - y^{(0)})$ from the associated $v'_i$ and an edge of weight $-(4y^{(1)} + 4y^{(0)})/(y^{(1)} - y^{(0)})$ from the constant vertex. These provide an overall input of at least $4$ to the new vertex if $v'_i$ has an output greater than $y^{(1)}$ and an input of at most $-4$ if $v'_i$ has an output less than $y^{(0)}$.

This ensures that if the outputs of the $v'_i$ encode binary values $x_1, ..., x_m$ appropriately, then each of the new vertices will output the value corresponding to the output of the appropriate gate or input. So, these vertices compute $h(x_1, ..., x_m)$ correctly. Furthermore, since the input to each of these vertices is outside of $[-3, 2]$, the derivatives of their activation functions with respect to their inputs are all $0$.

---

[12]Note that these will not be the $n$ data input of the general neural net that is being built; these input vertices take both the data inputs and some inputs from the memory component.

As such, the derivative of the loss function with respect to any of the edges leading to them is always 0, and paths through them do not contribute to changes in the weights of edges leading to the $v_i'$. $\quad\square$

Our next order of business is to show that we can perform queries successfully. So, we define the query subgraph as follows:

**Definition 2.** *Given $\tau > 0$, let $Q$ be the weighted directed graph with vertices $v_0$, $v_1$, $v_2$, $v_2'$, $v_3$, $v_4$, $v_c$, and $v_i^r$ for $0 \le i < \log_2(1/\tau)$ and the following edges:*

1. *An edge of weight $1/12$ from $v_0$ to $v_1$*

2. *Edges of weight $1$ from $v_1$ to $v_2$ and $v_2'$, and from $v_2$ and $v_2'$ to $v_3$.*

3. *An edge of weight $1/4$ from $v_3$ to $v_4$.*

4. *An edge of weight $10$ from $v_c$ to $v_2$.*

5. *An edge of weight $-10$ from $v_c$ to $v_2'$.*

6. *An edge of weight $-1/4$ from $v_c$ to $v_3$.*

7. *An edge of weight $12/\tau$ from $v_1$ to $v_i^r$ for each $i$.*

8. *An edge of weight $-1/\tau + 6 - 6 \cdot 2^{\lceil \log_2(1/\tau) \rceil}$ from $v_0$ to $v_i^r$ for each $i$.*

9. *An edge of weight $-3 \cdot 2^i$ from $v_i^r$ to $v_j^r$ for each $i > j$.*

*Also, let $Q'$ be the graph that is exactly like $Q$ except that in it the edge from $v_3$ to $v_4$ has a weight of $-1$.*

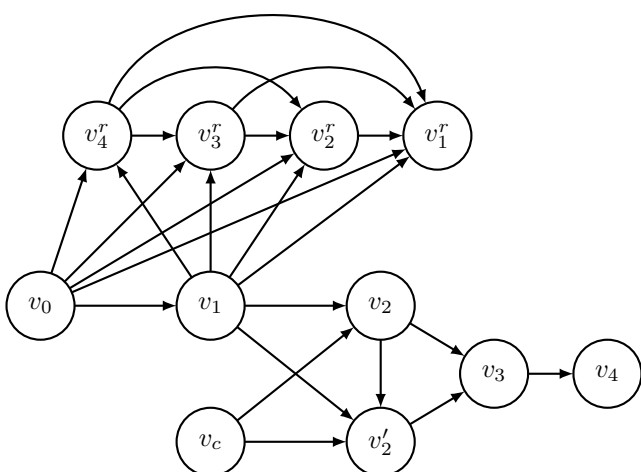

The idea behind this construction is as follows. $v_0$ will be the constant vertex, and $v_4$ will be the output vertex. If $v_c$ has activation $2$ then $v_2$ will have activation $2$ and $v_2'$ will have activation $-2$, leaving $v_3$ with activation $0$. So, this subgraph will have no effect on the output and the derivative of the loss with respect to any of its edge weights will be $0$. However, if $v_c$ has activation $0$ then this subgraph will contribute to the net's output, and the weights of the edges will change. So, when we do not want to use this subgraph to perform a query we simply set $v_c$ to $2$. In a timestep where we do want to use it to perform a query, we set $v_c$ to $0$ or $2$ based on the query's value on the current input so that the weight of the edge from $v_0$ to $v_1$ will change based on the value of the query. The activations of the $v^r$ will always give a binary representation of the activation of $v_1$. So, we can read off the current weight of the edge from $v_0$ to $v_1$. This construction works in the following sense.

**Lemma 11** (Editing memory). *Let $b$ be an integer greater than $1$, and $0 < \tau < 1/12$. Next, let $(f, G)$ be a neural net such that $G$ contains $Q$ or $Q'$ as a subgraph with $v_0$ as the constant vertex and $v_4$ as $G$'s output vertex, and there are no edges from vertices outside this subgraph to vertices in the subgraph other than $v_c$ and $v_4$. Now, assume that this neural net is trained using bSGD with learning rate $2$ and loss function $L$ for $T$ time steps, and the following hold:*

1. $v_c$ *outputs* $0$ *or* $2$ *on every sample in every time step.*

2. *The sample output is always* $\pm 1$.

3. *There is at most one timestep in which the net has a sample on which* $v_c$ *outputs* $0$. *On any such sample, the output of the net is* $-2\tau$ *if the subgraph is* $Q$ *and* $1 + 2\tau$ *if the subgraph is* $Q'$.

4. *The derivatives of the loss function with respect to the weights of all edges leaving this subgraph are always* $0$.

*The edge from* $v_3$ *to* $v_4$ *makes no contribution to* $v_4$ *on any sample where* $v_c$ *is* $2$, *a contribution of exactly* $1/24$ *on any sample where* $v_c$ *is* $0$ *if the subgraph is* $Q$, *and a contribution of exactly* $-1/24$ *on any sample where* $v_c$ *is* $0$ *if the subgraph is* $-Q$. *Also, if we regard an output of* $2$ *as representing the digit* $1$ *and an output of* $-2$ *as representing the digit* $0$ *then the binary string formed by concatenating the outputs of* $v^r_{\lceil \log_2(1/\tau) \rceil - 1}, ..., v^r_0$ *is within* $3/2$ *of the number of samples in previous steps where* $v_c$ *output* $0$ *and the net's output did not match the sample output divided by* $b\tau$ *plus double the total number of samples in previous steps where* $v_c$ *output* $0$ *divided by* $b$.

*Proof.* First, let let $t$ be the timestep on which there is a sample with $v_c$ outputting $0$ if any, and $T + 1$ otherwise. We claim that none of the weights of edges in this subgraph change on any timestep except step $t$, and prove it by induction on $t$. First, observe that by the definition of $t$, given any sample the net receives on a timestep before $t$, $v_c$ has an output of $2$. So, on any such sample $v_2$ and $v'_2$ will output $2$ and $-2$ respectively, which results in $v_3$ having an input of $-1/2$ and output of $0$, and thus in both the derivative of $v_4$ with respect to the weight of the edge from $v_3$ to $v_4$ and the derivative of $v_4$ with respect to the input of $v_3$ being $0$. So, the derivative of the loss with respect to any of the edge weights in this component is $0$ on all samples received before step $t$.

During step $t$, for any sample on which $v_c$ has an output of $0$, $v_1$, $v_2$, and $v'_2$ all have output $1/12$. That results in $v_3$ having an output of $1/6$. So, the derivative of the loss with respect to the weight of the edge from $v_0$ to $v_1$ is $\tau$ if the net's output agrees with the sample output and $(1 + 2\tau)/2$ otherwise. That means that the weight of the edge from $v_0$ to $v_1$ increases by $1/b$ times the number of samples in this step for which $v_c$ had output $0$ and the net's output disagreed with the sample output plus $2\tau/b$ times the total number of samples in this step for which $v_c$ had an output of $0$ plus an error term of size at most $3\tau/2$. Meanwhile, all of the other edges in the subgraph change by at most $1 + 3\tau$, and the weights of the edges from $v_2$ and $v'_2$ to $v_3$ are left with weights within $3\tau$ of each other because the derivatives of the gradients with respect to their weights are the same by symmetry, so only the error term differentiates them. The weights of the edges from $v_c$ do not change because the samples for which this subgraph's gradient is nonzero all have $v_c$ outputting $0$ and thus making their weights irrelevant.

On any step after step $t$, $v_c$ gives an output of $2$ on every sample. The weights of the edges from $v_1$ to $v_2$ and $v'_2$ have absolute value of at most $3$, so the edges from $v_c$ still provide enough input to them to ensure that they output $2$ and $-2$ respectively. That means that the input to $v_3$ is within $6\tau$ of $-1/2$, and thus that $v_3$ outputs $0$. That in turn means that the derivative of the loss with respect to any of the edge weights in the subgraph are $0$ so their weights do not change. All of this combined shows that the edge weights only ever change in step $t$ as desired.

Now, define even $m$ such that the weight of the edge from $v_0$ to $v_1$ increased by $m\tau$ in step $t$. We know that $m\tau$ is within $3\tau/2$ of the fraction of samples in step $t$ for which $v_c$ output $0$ and the net's output did not match the sample output plus $\tau$ times the overall fraction of samples in step $t$ for which $v_c$ output $0$. The output of $v_1$ is $1/12$ until step $t$ and $m\tau + 1/12$ after step $t$. Also, let $k = \lceil \log_2(1/\tau) \rceil$. Now, pick some $t' > t$ and let $r_i$ be the output of $v^r_i$ on step $t'$. In order to prove that the $r_i$ will be a binary encoding of $m$ we induct on $k - i$. So, assume that $v^r_{i+1}, ..., v^r_{k-1}$ encode

the correct binary digits. Then the input to $v_i^r$ is

$$(12/\tau)(m\tau + 1/12) - 1/\tau + 6 - 6 \cdot 2^k - \sum_{j=i+1}^{k-1} 3 \cdot 2^j r_j$$

$$= 12m + 6 - 6 \cdot 2^k + \sum_{j=i+1}^{k-1} 6 \cdot 2^j - \sum_{j=i+1}^{k-1} 12 \cdot 2^j \frac{r_j + 2}{4}$$

$$= 12m + 6 - 6 \cdot 2^{i+1} - 12 \cdot 2^{i+1} \lfloor m/2^{i+1} \rfloor$$

$$= 12 \left( m - 2^{i+1} \lfloor m/2^{i+1} \rfloor - 2^i + 1/2 \right)$$

If the $2^i$ digit of $m$'s binary representation is 1 this will be at least 6 while if it is 0 it will be at most $-6$, so $v_i^r$ will output 2 if the digit is 1 and $-2$ if it is 0 as desired. Showing that they all output $-2$ before step $t$ is just the $m = 0$ case of this.

Finally, recall that on every sample received after step $t$, $v_c$ will output 2, $v_3$ will output 0, and thus the edge from $v_3$ to $v_4$ will provide no input to $v_4$. During or before step $t$, the edges in this subgraph will all still have their original weights. So, if $v_c$ outputs 2 then $v_3$ outputs 0 and makes no contribution to $v_4$, while if $v_c$ outputs 0 then $v_3$ outputs $1/6$ and makes a contribution of magnitude $1/24$ and the appropriate sign to $v_4$, as desired. $\qquad \square$

*Proof of Lemma 3b.* At this point, we claim that we can build a neural net that emulates any $\mathsf{bSQ}_{\mathsf{TM}}^{0/1}$ algorithm using the same value of $b$ and an error of $\tau/4$, provided $\tau < 1/3$. In order to do that, we will structure our net as follows. First of all, we will have $(p \log_2(1/\tau) + 3)T$ copies of $Q$ and $Q'$. Then, we build a computation component that takes input from all the copies of the $v^r$ and computes from them what to output in the next step, what to query next, and what values those queries take on the current input. This component is built never to change as explained in Lemma 10. We will use a loss function of $L$ and learning rate of 2 when we train this net.

The net will also have $T$ primary output control vertices, $(p \log_2(1/\tau) + 3)T$ secondary output control vertices, and 1 final output control vertex. Each of these will have edges of weight 1 from two different outputs of the computation component and an edge of weight $-1/2$ from the constant vertex. That way, the computation component will be able to control whether each of these vertices outputs $-2$, 0, or 2. Each primary output control vertex will have an edge of weight $(1 + 2\rho)/2$ to the output, each secondary output control vertex will have an edge of weight $1/48$ to the output, and the final output control vertex will have an edge of weight $1/2$ to the output. Our plan is to set a new group of output control vertices to nonzero values in each timestep and to use it to control the net's output.

Each copy of $v_c$ will have an edge of weight $1/2$ from an output of the computation component and an edge of weight 1 from the constant vertex so the computation component will be able to control if it outputs 0 or 2. That allows the computation component to query an arbitrary function to $\{0, 1\}$ that is 0 whenever the sample output takes on the wrong value by setting $v_c$ to 0 on every input for which the function is potentially nonzero and 1 on every input on which it is 0 regardless of the sample output. Then the computation component can use a primary output control vertex to provide a value of $\pm(1 + 2\rho)$ to the output and use a set of secondary output control vertices to cancel out the effects of the copies of $Q$ and $Q'$ on the output.

A little more precisely, we will have one primary output control vertex, $(p \log_2(1/\tau) + 3)$ secondary output control vertices, and $(p \log_2(1/\tau) + 3)$ copies of $Q$ and $Q'$ associated with each time step. Then, in that step it will determine what queries the $\mathsf{bSQ}_{\mathsf{TM}}^{0/1}$ algorithm it is emulating would have made, and use the copies of $Q$ and $Q'$ to query the first $\lfloor \log_2(1/\tau) \rfloor + 2$ digits of each of them and the constant function 1. In order to determine the details of this, it will consider the current time step as being the first step for which the component querying the constant function still gives an output of 0 and consider each prior query performed by the $\mathsf{bSQ}_{\mathsf{TM}}^{0/1}$ algorithm as having given an output equal to the sum over $1 \leq i \leq \log_2(1/\tau) + 2$ of $\rho/2^i$ times the value given by the copy of $Q$ or $Q'$ used to query its $i$th bit. For any given input/output pair the actual value of the query will be within $\rho/2$ of the value given by its first $\lfloor \log_2(1/\tau) \rfloor + 2$ binary digits. Also, if the conditions

of Lemma 11 are satisfied then the value derived from the outputs of the $Q$ and $Q'$ will be within $\sum_i (7/2)\rho \cdot 2^{-i} \leq (7/2)\rho$ of the average of the values of the first $\lfloor \log_2(1/\tau) \rfloor + 2$ bits of the queried function on the batch. So, the values used by the computation component will be within $\tau$ of the average values of the queried functions on the appropriate batches, as desired. Also, any component that was ever used to query the function 1 will always return a nonzero value, so the computation component will be able to track the current timestep correctly.

We claim that in every timestep the net will output $-2\rho$ or $1 + 2\rho$ as determined by the computation component, the query subgraphs will update so that the computation component can read the results of the desired queries from them, and none of the edges will change in weight except the appropriate edges in copies of $Q$ or $Q'$ and possibly weights of edges from the output control vertices used in this timestep to the output, and we can prove this by induction on the timesteps.

So, assume that this has held so far. In the current timestep all of the output control vertices except those designated for this timestep are set to 0, and the ones designated for this timestep are set to the value chosen by the computation component. The weights of the edges from the current output control vertices still have their original values because the only way they could not is if they had been set to nonzero values before. The edges from the other output control vertices to the output have no effect on its value so the primary output control vertices as a whole make a contribution of $\pm(1 + 2\rho)$ as chosen by the computation component to the output. Meanwhile, there at most $(p \log_2(1/\tau) + 3)$ copies of $Q$ or $Q'$ that have $v_c$ set to 0 for any sample, so the computation component can compute the contribution they make to the output and use the secondary output control vertices for the timestep to cancel it out. So, the input to the output vertex will be exactly what we wanted it to be, which means that the copies of $Q$ and $Q'$ will update in the manner given by Lemma 11. That in turn means that the query subgraphs will update in the desired manner and the computation component will be able to determine valid values for the queries the $\mathsf{bSQ}_{\mathsf{TM}}^{0/1}$ algorithm would have made. None of the edges in or to the computation component will change by Lemma 10, and none of the edges from the computation component to any of the copies of $v_c$ or any output control vertices will change because they are always at flat parts of their activation functions. So, the net behaves as described. That means that the net continues to be able to make queries, perform arbitrary efficient computations on the results of those queries, and output the result of an arbitrary efficient computation.

Once it is done training the compuation component can use the final output control vertex to make the net output 0 or 1 based on the current input and the valuesof the previous queries. This process takes $k$ steps to run, and uses a polynomial number of query subgraphs per step. Any Turing machine can be converted into a circuit with size polynomial in the number of steps it runs for, so this can all be done by a net of size polynomial in the parameters. $\qquad\square$

## D  bSGD versus PAC and SQ : Proofs of Theorems 1a to 1d

Before proving Theorems 1a to 1d, we first prove Lemma 4, restated below for convenience.

**Lemma 4.** (bSGD to bSQ) *For all $T, \rho, b, p, r$, it holds that*

$$\mathsf{bSQ}\left(k = T, \tau = \frac{\rho}{4}, b, p, r\right) \preceq_0 \mathsf{bSGD}(T, \rho, b, p, r).$$

*Furthermore, for every poly-time computable activation $\sigma$, it holds that*

$$\mathsf{bSQ_{TM}}\left(k = T, \tau = \frac{\rho}{4}, b, p, r, \mathrm{TIME} = \mathrm{poly}(T, p, b, r)\right) \preceq_0 \mathsf{bSGD_{NN}^\sigma}(T, \rho, b, p, r).$$

*Proof.* Let $f_{\boldsymbol{w}}$ be some differentiable model. It suffices to show that a $\tau$-approximate rounding of the clipped gradient $\overline{\nabla}\mathcal{L}_S(f_{\boldsymbol{w}})$ can be guaranteed by querying a $(\tau/4)$-precision bSQ oracle. Indeed, we issue a query $\Phi : \mathcal{X} \times \mathcal{Y} \to [-1, 1]^p$ where $\Phi(x, y) = [\nabla \ell(f_{\boldsymbol{w}}(x), y)]_1$, and get some response $v$ satisfying $\|v - \mathbb{E}_S \nabla \ell(f_{\boldsymbol{w}}(x), y)\|_\infty \leq \tau/4$. Finally, observe that $g := \langle v \rangle_\tau := \arg\min_{v' \in (\tau \mathbb{Z})^p} \|v - v'\|_\infty$ is a $\tau$-approximate rounding of the required gradient. Indeed:

$$\|g - \nabla \mathcal{L}_S(f_{\boldsymbol{w}})\|_\infty \ \leq \ \|g - v\|_\infty + \|v - \nabla \mathcal{L}_S(f_{\boldsymbol{w}})\|_\infty \ \leq \ \tau/2 + \tau/4 \leq 3\tau/4. \qquad\square$$

Using the tools developed in Sections 4 and 5 we now prove Theorems 1a to 1d. We only show the proofs for the computationally unbounded case, as a near identical derivation achieves the required results for the computationally bounded case, since all relevant Theorems/Lemmas have both versions (except Lemmas 3a and 3b which are stated as different lemmas).

*Proof of Theorem 1a.* From Lemma 1, we have that for all $b$ and $\tau < 1/(2b)$, and $k = O(mn/\delta)$, $p = n + 1$ and $r' = r + k \log_2 b$, it holds that

$$\mathsf{bSQ}^{0/1}(k, \tau, b, p, r') \preceq_{\delta/2} \mathsf{PAC}(m, r)$$

From Lemma 3a (correspondingly Lemma 3b for the computationally bounded case), we then have that for $T = k = O(mn/\delta)$, $\rho = \tau/4 < 1/(8b)$, $p' = r' + (p+1)k = r + O((n + \log b)mn/\delta)$, it holds that

$$\mathsf{bSGD}(T, \rho, b, p, r') \preceq_{\delta/2} \mathsf{bSQ}^{0/1}(k, \tau, b, p, r')$$

The proof is complete by combining the above. $\square$

*Proof of Theorem 1b.* Any $\mathsf{bSGD}(T, \rho, b, p, r)$ algorithm can be simulated by a $\mathsf{PAC}(m, r)$ algorithm with $m = Tb$ samples, since $b$ samples are required to perform one bSGD iteration. Moreover, there is no loss in the error ensured by this simulation. $\square$

*Proof of Theorem 1c.* From Lemma 4 it holds for all $T, \rho, b, p, r$ and $k = T$, $\tau = \rho/4$ that

$$\mathsf{bSQ}(k, \tau, b, p, r) \preceq_0 \mathsf{bSGD}(T, \rho, b, p, r).$$

And by Theorem 2c, there exists a constant $C$ such that for $k' = kp = Tp$ and $\tau' = \frac{\tau}{2} = \frac{\rho}{8}$ it holds for all $b$ such that $b\tau^2 > C \log(kp/\delta)$ that

$$\mathsf{SQ}(k', \tau', r) \preceq_\delta \mathsf{bSQ}(k, \tau, b, p, r).$$

The proof is complete by combining the above. $\square$

*Proof of Theorem 1d.* From Lemma 2, it holds for all $b$, $\tau' = \frac{\tau}{4}$, $k' = k \cdot \left\lceil \frac{C \log(k/\delta)}{b\tau^2} \right\rceil$ and $p = 1$ that

$$\mathsf{bSQ}^{0/1}(k', \tau', b, p, r) \preceq_\delta \mathsf{SQ}(k, \tau, r)$$

Finally, from Lemma 3a (correspondingly Lemma 3b for the computationally bounded case) we have that for $T = k$, $\rho = \frac{\tau'}{4}$ and $p' = r + (p+1)k' = r + 2k'$ that

$$\mathsf{bSGD}(T, \rho, b, p', r) \preceq_0 \mathsf{bSQ}^{0/1}(k', \tau', b, p, r)$$

The proof is complete by combining the above. $\square$

## E  fbGD versus PAC and SQ : Proofs of Theorems 3a to 3d

Before proving Theorems 3a to 3d, we state the analogs of Lemmas 3a, 3b and 4 relating fbGD and fbSQ (fbSQ$^{0/1}$) (the proofs follow in an identical manner, so we skip it).

**Lemma 12a.** (fbSQ$^{0/1}$ to fbGD) *For all $\tau \in (0,1)$ and $m, k, p, r \in \mathbb{N}$, it holds that*

$$\mathsf{fbGD}\left(T' = k, \rho = \frac{\tau}{4}, m, p' = r + (p+1)k, r' = r\right) \preceq_0 \mathsf{fbSQ}^{0/1}(k, \tau, m, p, r).$$

**Lemma 12b.** (fbSQ$^{0/1}_{\mathsf{TM}}$ to fbGD$^\sigma_{\mathsf{NN}}$) *For all $\tau \in (0, \frac{1}{3})$ and $m, k, p, r \in \mathbb{N}$, and using the activation $\sigma$ from Figure 1, it holds that*

$$\mathsf{fbGD}^\sigma_{\mathsf{NN}}\left(T' = k, \rho = \frac{\tau}{4}, m, p' = \mathrm{poly}(k, \frac{1}{\tau}, m, p, r, \mathrm{TIME}), r' = r\right) \preceq_0 \mathsf{fbSQ}^{0/1}_{\mathsf{TM}}(k, \tau, m, p, r, \mathrm{TIME}).$$

**Lemma 13.** (fbGD to fbSQ) *For all $T, \rho, m, p, r$, it holds that*

$$\mathsf{fbSQ}\left(k = T, \tau = \frac{\rho}{4}, m, p, r\right) \preceq_0 \mathsf{fbGD}(T, \rho, m, p, r).$$

*Furthermore, for every poly-time computable activation $\sigma$, it holds for $\mathrm{TIME} = \mathrm{poly}(T, p, m, r)$ that*

$$\mathsf{fbSQ}_{\mathsf{TM}}\left(k = T, \tau = \frac{\rho}{4}, m, p, r, \mathrm{TIME} = \mathrm{poly}(T, p, m, r)\right) \preceq_0 \mathsf{fbGD}^\sigma_{\mathsf{NN}}(T, \rho, m, p, r).$$

Finally, we put together all the tools developed in Sections 4 and 5 along with the above Lemmas to prove Theorems 3a to 3d. As in Appendix D, we only show the proofs for the computationally unbounded case, as a near identical derivation achieves the required results for the computationally bounded case.

*Proof of Theorem 3a.* From Lemma 6, we have that for all $m$, $\tau < 1/(2m)$ and $r$, it holds for $k = 2m(n+1)$, $p = 1$, $r' = r$ that

$$\mathsf{fbSQ}^{0/1}(k, \tau, m, p, r') \preceq_0 \mathsf{PAC}(m, r)$$

From Lemma 12a (correspondingly Lemma 12b for the computationally bounded case), we then have that for $T = k = O(mn)$, $\rho = \tau/4 < 1/(8m)$, $p' = r' + (p+1)k = r + O(mn)$, it holds that

$$\mathsf{fbGD}(T, \rho, m, p', r') \preceq_\delta \mathsf{fbSQ}^{0/1}(k, \tau, m, p, r')$$

The proof is complete by combining the above. $\qquad\square$

*Proof of Theorem 3b.* Any $\mathsf{fbGD}(T, \rho, m, p, r)$ algorithm can be simulated by a $\mathsf{PAC}(m, r)$ algorithm with $m$ samples. Moreover, there is no loss in the error ensured by this simulation. $\qquad\square$

*Proof of Theorem 3c.* From Lemma 13 it holds for all $T, \rho, m, p, r$ and $k = T$, $\tau = \rho/4$ that

$$\mathsf{fbSQ}(k, \tau, m, p, r) \preceq_0 \mathsf{fbGD}(T, \rho, m, p, r).$$

And by Lemma 5c, there exists a constant $C$ such that for $k' = kp = Tp$ and $\tau' = \frac{\tau}{2} = \frac{\rho}{8}$ it holds for all $m$ such that $m\tau^2 > C(kp\log(1/\tau) + \log(1/\delta))$ that

$$\mathsf{SQ}(k', \tau', r) \preceq_\delta \mathsf{bSQ}(k, \tau, m, p, r).$$

The proof is complete by combining the above. $\qquad\square$

*Proof of Theorem 3d.* From Lemma 7, there exists a constant $C$ such that for all $m$, $\tau' = \frac{\tau}{4}$ satisfying $m\tau^2 \geq C(k\log(1/\tau) + \log(1/\delta))$ it holds for $k' = 2k$ and $p = 1$ that

$$\mathsf{fbSQ}^{0/1}(k', \tau', m, p, r) \preceq_\delta \mathsf{SQ}(k, \tau, r)$$

Finally, from Lemma 12a (correspondingly Lemma 12b for the computationally bounded case) we have that for $T = k'$, $\rho = \frac{\tau'}{4}$ and $p' = r + (p+1)k' = r + 2k'$ that

$$\mathsf{fbGD}(T, \rho, m, p', r) \preceq_0 \mathsf{fbSQ}^{0/1}(k', \tau', m, p, r)$$

The proof is complete by combining the above. $\qquad\square$