# OpenReview forum: "On the Power of Differentiable Learning versus PAC and SQ Learning"
_NeurIPS.cc/2021/Conference — NeurIPS 2021 Spotlight_

### Official Review · Reviewer_Gdae · 2021-07-04

**Rating:** 7
**Confidence:** 3

**Summary:**

The authors characterize the power of minibatch SGD on differentiable models in comparison to SQ and "PAC" (learners with access to samples) models.  They build upon previous work of Abbe & Sandon who had shown that online SGD on neural networks (batch size = 1) is 'universal' in the sense that if some function can be learned using samples in poly time, it can be learned using online SGD over some appropriate neural network with all relevant parameters polynomial, and further that this does not hold with full-batch GD.  The authors sharpen these results to precisely characterize the relationship between batch size/gradient noise using SGD on differentiable models and the competitiveness of this with (a) the class of "PAC" algorithms and (b) the class of SQ algorithms.  To do so they introduce the notion of batch SQ algorithms and analyze when inclusions between the different classes are strict or are equalities.


**Limitations And Societal Impact:**

Yes, they adequately discussed limitations

**Main Review:**


The results presented are interesting: understanding separations between SGD on differentiable models on the one hand, and SQ algorithms on the other, is an important topic.  A handful of recent works over the past two years have demonstrated separations between SGD and full-batch GD, leading to the reasonable conjecture that there is some threshold of "small batch" such that sufficient randomness (small batch) can enable universality while too much averaging (large batch) can lead to equivalencies with SQ.  (The authors should comment on the recent work [1'] below which also touches on this topic.)  The authors' results are a timely work in the context of the previous work, and develops a novel resolution of these conjectures that are either implicit or explicit in previous work.

The work is mostly clear, but there are some clarifications that I did not understand on my first and second reads, and given the tight turnaround time for reviews did not have time to examine them more carefully (nor to carefully read proofs in appendix).

(1) The authors' results are for binary input and binary output, suggesting classification problems, yet the loss considered is the squared loss.  Can they comment on this?  Is it possible to extend results to other loss functions?  Presumably not the 0-1 loss?

(2) In Theorem 1, it appears the time T' needed to run SGD on NNs does not depend on the time "TIME" needed for a PAC learner, neither on rho -- only the size of the network depends on TIME and rho.  Is this correct? Can you give intuition for this?

(3) The paper would be improved with a paragraph or two dedicated to clearly explaining the difference between rho approximate gradients, the typical tau used in SQ models, and the batch size used to estimate the gradients.  There are implicit equivalences given in e.g. Theorems 1 and 2, but having a paragraph where these settings are explicitly discussed would allow for readers more familiar with the standard concepts of batch size/concentration and the threshold in SQ algorithms to have a more intuitive grasp of this.


_Minor points of clarification._

(m1) Line 74 it is implied that "computationally unbounded SQ" is limited only by the number of statistical queries, but wouldn't it instead be bounded by the threshold tau / precision of the estimates?

(m2) in line 144 you mention 'r bits of randomness', in the context of X\in {0,1}^n should r refer to n+1 then?  Why the usage of r here?

(m3) line 323-324 "as to what makes a learning problem are learnable"

[1'] The Connection Between Approximation, Depth Separation and Learnability in Neural Networks. Eran Malach, Gilad Yehudai, Shai Shale-Shwartz, Ohad Shamir


** Post rebuttal **

I've read the author response and heartily recommend acceptance.


**Time Spent Reviewing:**

5

---

> ### Author Response · Authors · 2021-08-10
> **Response to Reviewer Gdae**
>
> We thank the reviewer for the very positive review and encouraging feedback. Below we answer the main questions raised by the reviewer:
> 1) Although as presented the simulation is specific to the squared loss, as it is easier to write down the simulation with a particular loss, it could have been done with essentially any "non-trivial enough" continuous loss.  The loss does have to be differentiable, or at least continuous and differentiable almost everywhere, so the 0/1 loss wouldn't work.  But the abs-loss or logistic or hinge loss are fine.  In fact, the simulation in Appendix E uses the absolute loss (we have since changed that too to the squared loss, for consistency with the rest of the paper, but this shows that it doesn’t matter much which loss we use).
> 2) Yes, that is correct. The number of iterations ($T’$) of SGD on NN does not depend on the TIME needed for a PAC learner. But note that the processing essentially happens in the forward propagation in the neural net, where the neural net simulates the computation of the PAC learner, so having a bigger net allows more complicated calculation with each batch. The size of the network (and hence the runtime of the forward propagation) does increase with TIME.
> 3) We agree that such discussion is in place, and will add it in a revision.
>     - At a technical level, a rho-approximate rounding of $v \in R$ can be obtained from a $\rho/4$ approximation $|v-\tilde{v}|<=\rho/4$, but rounding $\tilde{v}$ to the nearest integer multiple of $\rho$.
>     - The way we think of SQ vs bSQ (and hence bSGD) is as follows: much of the motivation for the tolerance tau in the standard definition of SQ learning, is to capture the fact that the query is answered based on a sample, since answering based on a sample of size b we will have accuracy O(1/sqrt(b)).  We can therefore ask what happens if we just take answers generated this way.  In a sense, we show that considering EXACT answers generated this way is too powerful, and we must also consider the numeric precision of the answers, which is captured by the bSQ parameter tau (or the bSGD parameter rho).  Our definition of bSQ can be thought of as decomposing the SQ tolerance to a sampling error O(1/sqrt(b)) and an arbitrary error (precision) tau.  With standard notions of computation and sampling, it is reasonable to consider working with poly many bits of precision, corresponding to exp small precision tau, and so a much smaller arbitrary error than sampling error.  One can ask if standard SQ learning is indeed a good surrogate for basing results on sampling+numeric error in this regime, i.e. when the arbitrary error is much smaller than the sampling error.  Our results indicate SQ does not capture this: that if the numerical error is much smaller (even poly smaller, let alone exp smaller) than the sampling error, bSQ becomes equivalent to PAC rather than to SQ.
>
> (m1) Correct, the computation is limited by the number _and precision_ of the queries, but not runtime.  We will fix this.
>
> (m2) Note that $r$ is the number of bits of randomness used internally by a randomized algorithm, and is not related to the input dimension (it is not about the randomness in the input sample). We will clarify this.
>
> (m3) We will fix this typo.

---

### Official Review · Reviewer_Sk3C · 2021-07-14

**Rating:** 7
**Confidence:** 2

**Summary:**

The authors show that how the power of SGD on differential models or neural networks with arbitrary initialization is affected by the relationship between mini-batch size and gradient estimate precision. Abbe and Sandon had already shown that SGD with mini-batch size 1 can simulate any poly-time learning algorithm, but that gradient descent is not more powerful than statistical query learning algorithms. This work gives a finer-grained dichotomy: that when gradient estimates have enough precision relative to batch size, then any learning algorithm can be simulated with a polynomially related number of samples, but for moderately lower precision gradient descent is equivalent in power to statistical query learning algorithms. For poly-time learning/SQ algorithms, the equivalences are shown for poly-size neural networks rather than general differential models.

**Limitations And Societal Impact:**

Yes, the authors provide an adequate discussion of the limitations of their work, especially in section 6, "summary and discussion".

**Main Review:**

Recent results by Abbe and Sandon, showing that SGD with batch size 1 on neural networks can simulate any PAC learning algorithm, have made it formally even more obvious than before that theoretical work on the capabilities and limitations of deep learning methods ought not to apply to all possible architectures and initializations, since the freedom to choose architecture and initialization arbitrarily is evidently so powerful. Their results also showed an intriguing gap between SGD and GD, GD being only as powerful as the weaker SQ learning. This paper paints a fuller picture, showing how the capabilities of SGD (on arbitrary network/model architectures with arbitrary initialization, both unrealistic) vary between PAC and SQ learning as batch size and gradient precision change. It is presented in an organized, readable way. The paper has no *practical* significance, as the authors readily admit, but rather helps clarify the landscape of SGD on unrestricted models/initialization in a way that might be useful for showing for which design choices future lower bounds ought not to apply.

I have not read the proofs carefully, nor have I read in depth the relevant works by Abbe and Sandon, so I am recommending acceptance with low confidence.

**Time Spent Reviewing:**

4

---

> ### Author Response · Authors · 2021-08-10
> **Answer to Reviewer Sk3C**
>
> Thanks for the positive and encouraging comments.

---

### Official Review · Reviewer_Pvqb · 2021-07-16

**Rating:** 7
**Confidence:** 3

**Summary:**

The paper studies the power of differentiable learning. Previous work of Abbe and Sandon has showed that any learning algorithm can be simulated using SGD with on a network that with a certain structure and certain initialization; furthermore, batch gradient descent was shown to be less powerful (less powerful than the statistical query model).

The current paper aims at extending this analysis to the case of batch gradient descent. It is assumed that the samples are encoded with a few bits (so not really continuous). The main result of the paper is that for small enough batches (and good precision on the batch-gradient) we can still simulate any algorithm [while this is not the case for larger batches].

**Ethics Review Area:**

["I don’t know"]

**Limitations And Societal Impact:**

I don't see any direct impact.

**Main Review:**

+ The authors use the term PAC learning but what they really mean is any algorithm that takes as input a training set and outputs a predictor. I don't think PAC is a good choice of terminology for this context. If I understand correctly the result suggest that any learning rule can be simulated.

+ The authors assume that the input samples are represented by a few bits (so not continuous). This should be made more clear throughout the paper.

+ An important part of the paper is to show that batch-statistical query model is as powerful as PAC. For this, the authors show how to extract samples using bSQ queries (but allowing multiple queries from the same batch). As part of Algorithm 1, we read "perform the batch statistical queries ... on an independently sampled batch...". This sounds like we generate different batches while we loop. However, I think all the queries in the loop should be performed on the same batch. Can the authors clarify if I am right? Otherwise I don't think the algorithm makes sense.

+ In case I was right about the above point, it means that the in the bSQ model we are not only allowed to ask multiple queries about the same batch, but also we can ask these queries adaptively. Does this affect your proof that bSGD is at least as powerful that bSQ (in the certain regime of b and rho)?

+ Although the authors talk about the work of Abbe and Sandon, they do not specifically say what is different in terms of the techniques used (I see that your results are new, but how is the proof techniques different?)

+ The model is somewhat peculiar since we create a huge network that looks really nothing like usual networks but when you perform bSGD on it, it emulates the desired algorithm.

-----------------------------------
My main questions has been answered by the authors in the rebuttal and I am increasing my score from 6 to 7.


**Time Spent Reviewing:**

8

---

> ### Author Response · Authors · 2021-08-10
> **Answer to Reviewer Pvqb**
>
> We thank the reviewer for the feedback and comments. Below we respond to the concerns and questions raised by the reviewer, including clarifying two issue (regarding not re-using the same sample, and regarding input representation in bit) that hopefully satisfy the reviewer regarding their major concerns:
> - On the validity of Algorithm 1: The algorithm is correct as written, namely - a fresh batch is used for each iteration in the algorithm. That is, to extract a single sample, we perform multiple rounds of bSQ queries (i.e. on multiple _independent_ mini-batches $S_t$), making $n+1$ parallel queries on each mini-batch $S_t$. The queries on a single minibatch are thus NOT allowed to be adaptive. We note that what Algorithm 1 does is generate a single example, sampled according to the (unknown) underlying distribution—-this is a bit different from extracting an example from a given subset of examples. We hope this explanation clarifies the purpose and the behavior of the proposed algorithm, and we will also try to clarify the text in the paper.
> - On the input being discrete: This is a good point, and following your comment we agree this is something we should (and will) discuss in the paper.  We indeed assume throughout the paper that the input is discrete and can be represented with a finite number of bits.  We feel this is sensible for the following reasons (which we will discuss in more detail in a revision):
>     - One should think of $n$ as the total number of bits in the input.  Even if the input is numerical/real-valued, if it is represented digitally, each number would be represented with some finite number of bits, and $n$ is the total number of bits.  And so, one can think of n as capturing the dimensionality and precision of the inputs.
>     - In particular, if considering time-bounded computation in a classical turing machine model (which we require in order to get equivalence with neural nets), the number of bits in the input needs to be finite and bounded by the runtime, and so this is not an issue. (One might consider a model of real-valued exact computation, but that would be impossible to simulate with the finite precision SGD we consider).
>     - When not considering computational constraints, if the input is real-valued, our simulation of arbitrary sample based methods would indeed depend on how fine a precision the method being simulated uses.  In particular, the model size and number of required SGD iterations would depend logarithmically on the precision use by the simulated method.  Although we find this a sensible (and likely unavoidable) dependence, since in other places we were careful about dependence on precision, it would be appropriate to mention this as well.
> - On the PAC terminology: We admit in footnote 5 (page 4) that our use of the acronym PAC is improper.  It does not correspond to the words the acronym represents, and it is used only as a tag for learning methods based on a training set of samples.  We use the tag “PAC” since over the decades we feel it has acquired this meaning, and in particular, by “The PAC Model” is frequently used to refer to “learning based on an iid sample of examples”.  As indicated in the footnote, we still feel uncomfortable with its use, and would be happy for a suggestion for an alternative short and evocative tag.
> - The technical difference from Abbe and Sandon (2020): while our work shares similar techniques with the work of Abbe and Sandon, we introduce new technical ideas that are essential for achieving the full strength of our results.
>     - The sample extraction mechanism, which is the core tool for simulating PAC with bSGD (going beyond batch-size of $1$), is entirely novel.
>     - The equivalence between bSGD and bSQ, in a fashion that is agnostic to the batch-size and precision of the gradients, is another core technique that has not been shown in previous works, and is essential for establishing our results.
>     - We show an almost tight characterization, in terms of the batch-size/precision tradeoff, of when bSGD can go beyond SQ learning.
>     - We show a simulation of generic differentiable models, which allows us to decouple the runtime of the SQ/PAC algorithms, and consider learning with both tractable and untractable methods.
>     - Our direct simulation of PAC/SQ using neural networks is similar to the one introduced by Abbe and Sandon, with some new elements that allow us to compute arbitrary queries.
>     - Our lower-bound is different than the one in Abbe-Sandon and relies on concentration bound rather than indistinguishability bounds from hypothesis testing.
>
> We hope that our response properly answered all the concerns. If so, please consider changing your score accordingly.  We would also be happy to further clarify if needed.

---

### Official Review · Reviewer_uqhv · 2021-07-20

**Rating:** 7
**Confidence:** 3

**Summary:**

This work studies the power of stochastic gradient descent with mini-batch (bSGD). The authors prove that bSGD is as power as SQ learning for all the batch size. Furthermore, they prove that there is a threshold $b'$ relative to the precision, such that if the batch size is less than $b'$ then bSGD is as powerful as PAC learning. Previous work i.e., [1] have studied the case of SGD. The authors define a new model bSQ and prove its computational power and then prove that bSGD is as powerful as bSQ.

**Limitations And Societal Impact:**

As a theory paper, it doesn't have any negative societal impact.

**Main Review:**

1. Well-written and organized.
2. This work establishes some fundamental results in PAC learning and SQ learning as well as on the power of bSGD algorithms.
3. Introduces a new model bSQ which can be of independent interest.
4. They show the computational time of the reduction is polynomial, i.e., it does not need a network of exponential size if the PAC learning algorithm has poly-time.

Overall, I think this is a good paper and I recommend acceptance.

#Other comments:
- $\mathcal{D}^b$ isn't defined anywhere or $S_t$ uniform over samples.
- Line 232: "Lemmas" to "theorems"


**Time Spent Reviewing:**

3-4

---

> ### Author Response · Authors · 2021-08-10
> **Answer to Reviewer uqhv**
>
> We thank the reviewer for the positive review and encouraging feedback. We will clarify that $\mathcal{D}^b$ is the product distribution corresponding to $b$ iid draws from $\mathcal{D}$; moreover, each $S_t$ is an an independent draw from $\mathcal{D}^b$.

---

### Decision · Program_Chairs · 2021-09-28

**Decision:**

Accept (Spotlight)

**Comment:**

This presents interesting results regarding the power of mini-batch SGD. All reviewers and the AC like the paper and the AC recommend acceptance.

**Consistency Experiment:**

NeurIPS has a long history of experimentation. In 2014, NeurIPS ran an experiment in which 10% of submissions were reviewed by two independent committees to quantify the randomness in the review process. This year, we repeated a variant of this experiment to see how the quality of the review process has changed over time.  This paper was part of the experiment and was therefore assigned to two committees (consisting of reviewers, an Area Chair, and a Senior Area Chair) that reached independent decisions.  If both committees made the same recommendation, this recommendation was followed. If a single committee recommended acceptance, the paper was accepted (with the exception of a few cases in which the other committee identified what we considered a fatal flaw, e.g., an error in a key result).

This copy’s committee reached the following decision: **Accept (Poster)**

The other committee assigned to the paper recommended **Accept (Spotlight)**.  You can find the other set of reviews, along with any follow up discussion with the authors here:
https://openreview.net/forum?id=TZPidZS3r_z